# Exact Solutions of a Deep Linear Network

**Liu Ziyin[1], Botao Li[2], Xiangming Meng[3]**
[1]Department of Physics, The University of Tokyo
[2]Laboratoire de Physique de l'Ecole normale supérieure, ENS, Université PSL,
CNRS, Sorbonne Université, Université de Paris Cité, Paris, France
[3]Institute for Physics of Intelligence, Graduate School of Science, The University of Tokyo

## Abstract

This work finds the analytical expression of the global minima of a deep linear network with weight decay and stochastic neurons, a fundamental model for understanding the landscape of neural networks. Our result implies that zero is a special point in deep neural network architecture. We show that weight decay strongly interacts with the model architecture and can create bad minima at zero in a network with more than 1 hidden layer, qualitatively different from a network with only 1 hidden layer. Practically, our result implies that common deep learning initialization methods are insufficient to ease the optimization of neural networks in general.

## 1 Introduction

Applications of neural networks have achieved great success in various fields. One central open question is why neural networks, being nonlinear and containing many saddle points and local minima, can sometimes be optimized easily (Choromanska et al., 2015a) while becoming difficult and requiring many tricks to train in some other scenarios (Glorot and Bengio, 2010; Gotmare et al., 2018). One established approach is to study the landscape of deep linear nets (Choromanska et al., 2015b), which are believed to approximate the landscape of a nonlinear net well. A series of works proved the famous results that for a deep linear net, all local minima are global (Kawaguchi, 2016; Lu and Kawaguchi, 2017; Laurent and Brecht, 2018), which is regarded to have successfully explained why deep neural networks are so easy to train because it implies that initialization in any attractive basin can reach the global minimum without much effort (Kawaguchi, 2016). However, the theoretical problem of when and why neural networks can be hard to train is understudied.

In this work, we theoretically study a deep linear net with weight decay and stochastic neurons, whose loss function takes the following form in general:

$$\underbrace{\mathbb{E}_x \mathbb{E}_{\epsilon^{(1)},\epsilon^{(2)},...,\epsilon^{(D)}} \left( \sum_{i,i_1,i_2,...,i_D}^{d,d_1,d_2,...d_D} U_{i_D} \epsilon_{i_D}^{(D)} ... \epsilon_{i_2}^{(2)} W_{i_2 i_1}^{(2)} \epsilon_{i_1}^{(1)} W_{i_1 i}^{(1)} x_i - y \right)^2}_{L_0} + \underbrace{\gamma_u \|U\|_2^2 + \sum_{i=1}^{D} \gamma_i \|W^{(i)}\|_F^2}_{L_2\ reg.},$$
(1)

where $\mathbb{E}_x$ denotes the expectation over the training set, $U$ and $W^{(i)}$ are the model parameters, $D$ is the depth of the network,[1] $\epsilon$ is the noise in the hidden layer (e.g., due to dropout), $d_i$ is the width of the $i$-th layer, and $\gamma$ is the strength of the weight decay. Previous works have studied special cases of this loss function. For example, Kawaguchi (2016) and Lu and Kawaguchi (2017) study the landscape of $L_0$ when $\epsilon$ is a constant (namely, when there is no noise). Mehta et al. (2021) studies

36th Conference on Neural Information Processing Systems (NeurIPS 2022).

---

[1]In this work, we use "depth" to refer to the number of hidden layers. For example, a linear regressor has depth 0.

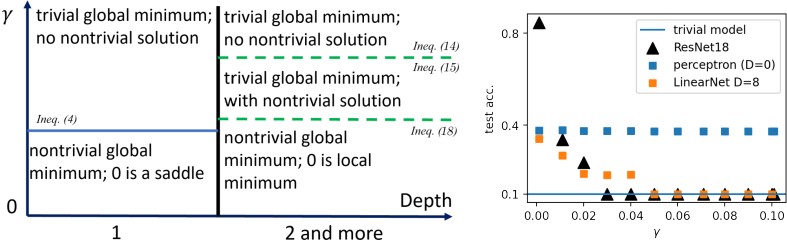

Figure 1: **Left**: A summary of the network landscape that is implied by the main results of this work when one increases the weight decay strength $\gamma$ while fixing other terms. We show that the landscape of a depth-1 net can be precisely divided into two regimes, while, for $D \geq 2$, there exists at least three regimes. The solid blue line indicates that the division of the regimes is precisely understood. The dashed lines indicate that the conditions we found are not tight and may be improved in the future. **Right**: ResNet18 on CIFAR10. The performance of a linear regressor never drops to that of a trivial model, whereas the performance of ResNet18 drops to the level of a trivial model, like a deep linear net with similar depth.

$L_0$ with (a more complicated type of) weight decay but without stochasticity and proved that all the stationary points are isolated. Another line of works studies $L_0$ when the noise is caused by dropout (Mianjy and Arora, 2019; Cavazza et al., 2018). Our setting is more general than the previous works in two respects. First, apart from the mean square error (MSE) loss $L_0$, an $L_2$ regularization term (weight decay) with arbitrary strength is included; second, the noise $\epsilon$ is arbitrary. Thus, our setting is arguably closer to the actual deep learning practice, where the injection of noises to latent layers is common, and the use of weight decay is virtually ubiquitous (Krogh and Hertz, 1992; Loshchilov and Hutter, 2017). One major limitation of our work is that we assume the label $y$ to be 1-dimensional, and it can be an important future problem to prove whether an exact solution exists or not when $y$ is high-dimensional.

Our foremost contribution is to prove that all the global minimum of an arbitrarily deep and wide linear net takes a simple analytical form. In other words, we identify in closed form the global minima of Eq. (1) up to a single scalar, whose analytical expression does not exist in general. We then show that it has nontrivial properties that can explain many phenomena in deep learning. In particular, the implications of our result include (but are not limited to):

1. Weight decay makes the landscape of neural nets more complicated;

   - we show that bad minima[2] emerge as weight decay is applied, whereas there is no bad minimum when there is no weight decay. This highlights the need to escape bad local minima in deep learning with weight decay.

2. Deeper nets are harder to optimize than shallower ones;

   - we show that a $D \geq 2$ linear net contains a bad minimum at zero, whereas a $D = 1$ net does not. This partially explains why deep networks are much harder to optimize than shallower ones in deep learning practice.

3. Depending on the task, the common initialization methods (such as the Kaiming init.) can initialize a deep model in the basin of attraction of the bad minimum at zero;

   - common initialization methods initialize the models at a radius of roughly $1/\sqrt{width}$ around the origin; however, we show that the width of the bad minimum is task-dependent and can be larger than the initialization radius for tasks with a small margin ($\|\mathbb{E}[xy]\|$);

4. Thus, the use of (effective) weight decay is a major cause of various types of collapses in deep learning (for example, see Figure 1).

**Organization**: In the next section, we discuss the related works. In Section 3, we derive the exact solution for a two-layer net. Section 4 extends the result to an arbitrary depth. In Section 5, we study and discuss the relevance of our results to many commonly encountered problems in deep learning. The last section concludes the work and discusses unresolved open problems. All proofs are delayed to Section B. Moreover, additional theoretical results on the effect of including a bias term is considered in Section D.

---

[2]Unless otherwise specified, we use the word "bad minimum" to mean a local minimum that is not a global minimum.

**Notation**. For a matrix $W$, we use $W_{i:}$ to denote the $i$-th row vector of $W$. $\|Z\|$ denotes the $L_2$ norm if $Z$ is a vector and the Frobenius norm if $Z$ is a matrix. The notation $*$ signals an optimized quantity. Additionally, we use the superscript $^*$ and subscript $_*$ interchangeably, whichever leads to a simpler expression. For example, $b_*^2$ and $(b^*)^2$ denote the same quantity, while the former is "simpler."

## 2    Related Works

In many ways, linear networks have been used to help understand nonlinear networks. For example, even at depth 0, where the linear net is just a linear regressor, linear nets are shown to be relevant for understanding the generalization behavior of modern overparametrized networks (Hastie et al., 2019). Saxe et al. (2013) studies the training dynamics of a depth-1 network and uses it to understand the dynamics of learning of nonlinear networks. These networks are the same as a linear regression model in terms of expressivity. However, the loss landscape is highly complicated due to the existence of more than one layer, and linear nets are widely believed to approximate the loss landscape of a nonlinear net (Kawaguchi, 2016; Hardt and Ma, 2016; Laurent and Brecht, 2018). In particular, the landscape of linear nets has been studied as early as 1989 in Baldi and Hornik (1989), which proposed the well-known conjecture that all local minima of a deep linear net are global. This conjecture is first proved in Kawaguchi (2016), and extended to other loss functions and deeper depths in Lu and Kawaguchi (2017) and Laurent and Brecht (2018). Many relevant contemporary deep learning problems can be understood with deep linear models. For example, two-layer linear VAE models are used to understand the cause of the posterior collapse problem (Lucas et al., 2019; Wang and Ziyin, 2022). Deep linear nets are also used to understand the neural collapse problem in contrastive learning (Tian, 2022). We also provide more empirical evidence in Section 5.

## 3    Two-layer Linear Net

This section finds the global minima of a two-layer linear net. The data point is a $d$-dimensional vector $x \in \mathbb{R}^d$ drawn from an arbitrary distribution, and the labels are generated through an arbitrary function $y = y(x) \in \mathbb{R}$. For generality, we let different layers have different strengths of weight decay even though they often take the same value in practice. We want to minimize the following objective:

$$L_{d,d_1}(U,W) = \mathbb{E}_x \mathbb{E}_\epsilon \left( \sum_j^{d_1} U_j \epsilon_j \sum_i^d W_{ji} x_i - y \right)^2 + \gamma_w \|W\|^2 + \gamma_u \|U\|^2, \qquad (2)$$

where $d_1$ is the width of the hidden layer and $\epsilon_i$ are independent random variables. $\gamma_w > 0$ and $\gamma_u > 0$ are the weight decay parameters. Here, we consider a general type of noise with $\mathbb{E}[\epsilon_i] = 1$ and $\mathbb{E}[\epsilon_i \epsilon_j] = \delta_{ij} \sigma^2 + 1$ where $\delta_{ij}$ is the Kronecker's delta, and $\sigma^2 > 0$.[3] For shorthand, we use the notation $A_0 := \mathbb{E}[xx^T]$, and the largest and the smallest eigenvalues of $A_0$ are denoted as $a_{\max}$ and $a_{\min}$ respectively. $a_i$ denotes the $i$-th eigenvalue of $A_0$ viewed in any order. For now, it is sufficient for us to assume that the global minimum of Eq. (2) always exists. We will prove a more general result in Proposition 1, when we deal with multilayer nets.

### 3.1    Main Result

We first present two lemmas showing that the global minimum can only lie on a rather restrictive subspace of all possible parameter settings due to invariances in the objective.

**Lemma 1.** *At the global minimum of Eq.* (2), $U_j^2 = \frac{\gamma_w}{\gamma_u} \sum_i W_{ji}^2$ *for all* $j$.

*Proof Sketch.* We use the fact that the first term of Eq. (2) is invariant to a simultaneous rescaling of rows of the weight matrix to find the optimal rescaling, which implies the lemma statement. □

This lemma implies that for all $j$, $|U_j|$ must be proportional to the norm of its corresponding row vector in $W$. This lemma means that using weight decay makes all layers of a deep neural network balanced. This lemma has been referred to as the "weight balancing" condition in recent works (Tanaka et al., 2020), and, in some sense, is a unique and potentially essential feature of neural networks that encourages a sparse solution (Ziyin and Wang, 2022). The following lemma further shows that, at the global minimum, all elements of $U$ must be equal.

---

[3]While we formally require $\gamma$ and $\sigma$ to nonzero, one can show that the solutions we provided remain global minimizers in the zero limit by applying Theorem 2 from Ziyin and Ueda (2022).

**Lemma 2.** *At the global minimum, for all $i$ and $j$, we have*

$$\begin{cases} U_i^2 = U_j^2; \\ U_i W_{i:} = U_j W_{j:}. \end{cases} \tag{3}$$

*Proof Sketch.* We show that if the condition is not satisfied, then an "averaging" transformation will strictly decrease the objective. □

This lemma can be seen as a formalization of the intuition suggested in the original dropout paper (Srivastava et al., 2014). Namely, using dropout encourages the neurons to be independent of one another and results in an averaging effect. The second lemma imposes strong conditions on the solution of the problem, and the essence of this lemma is the reduction of the original problem to a lower dimension. We are now ready to prove our first main result.

**Theorem 1.** *The global minimum $U_*$ and $W_*$ of Eq. (2) is $U_* = 0$ and $W_* = 0$ if and only if*

$$\|\mathbb{E}[xy]\|^2 \le \gamma_u \gamma_w. \tag{4}$$

*When $\|\mathbb{E}[xy]\|^2 > \gamma_u \gamma_w$, the global minima are*

$$\begin{cases} U_* = b\mathbf{r}; \\ W_* = \mathbf{r}\mathbb{E}[xy]^T b \left[ b^2 \left( \sigma^2 + d_1 \right) A_0 + \gamma_w I \right]^{-1}, \end{cases} \tag{5}$$

*where $\mathbf{r} = (\pm 1, ..., \pm 1)$ is an arbitrary vertex of a $d_1$-dimensional hypercube, and $b$ satisfies:*

$$\left\| \left[ b^2 \left( \sigma^2 + d_1 \right) A_0 + \gamma_w I \right]^{-1} \mathbb{E}[xy] \right\|^2 = \frac{\gamma_u}{\gamma_w}. \tag{6}$$

Apparently, $b = 0$ is the trivial solution that has not learned any feature due to overregularization. Henceforth, we refer to this solution (and similar solutions for deeper nets) as the "trivial" solution. We now analyze the properties of the nontrivial solution $b^*$ when it exists.

The condition for the solution to become nontrivial is interesting: $\|\mathbb{E}[xy]\|^2 \ge \gamma_u \gamma_w$. The term $\|\mathbb{E}[xy]\|$ can be seen as the effective strength of the signal, and $\gamma_u \gamma_w$ is the strength of regularization. This precise condition means that the learning of a two-layer can be divided into two qualitatively different regimes: an "overregularized regime" where the global minimum is trivial, and a "feature learning regime" where the global minimum involves actual learning. Lastly, note that our main result does not specify the exact value of $b^*$. This is because $b^*$ must satisfy the condition in Eq. (6), which is equivalent to a high-order polynomial in $b$ with coefficients being general functions of the eigenvalues of $A_0$, whose solutions are generally not analytical by Galois theory. One special case where an analytical formula exists for $b$ is when $A_0 = \sigma_x^2 I$. See Section C for more discussion.

### 3.2   Bounding the General Solution

While the solution to $b^*$ does not admit an analytical form for a general $A_0$, one can find meaningful lower and upper bounds to $b^*$ such that we can perform an asymptotic analysis of $b^*$. At the global minimum, the following inequality holds:

$$\| \left[ b^2 \left( \sigma^2 + d_1 \right) a_{\max} I + \gamma_w I \right]^{-1} \mathbb{E}[xy] \|^2 \le \| \left[ b^2 \left( \sigma^2 + d_1 \right) A_0 + \gamma_w I \right]^{-1} \mathbb{E}[xy] \|^2$$
$$\le \| \left[ b^2 \left( \sigma^2 + d_1 \right) a_{\min} I + \gamma_w I \right]^{-1} \mathbb{E}[xy] \|^2, \tag{7}$$

where $a_{\min}$ and $a_{\max}$ are the smallest and largest eigenvalue of $A_0$, respectively. The middle term is equal to $\gamma_u / \gamma_w$ by the global minimum condition in (33), and so, assuming $a_{\min} > 0$, this inequality is equivalent to the following inequality of $b^*$:

$$\frac{\sqrt{\frac{\gamma_w}{\gamma_u}} \|\mathbb{E}[xy]\| - \gamma_w}{(\sigma^2 + d_1) a_{\max}} \le b_*^2 \le \frac{\sqrt{\frac{\gamma_w}{\gamma_u}} \|\mathbb{E}[xy]\| - \gamma_w}{(\sigma^2 + d_1) a_{\min}}. \tag{8}$$

Namely, the general solution $b^*$ should scale similarly to the homogeneous solution in Eq. (105) if we treat the eigenvalues of $A_0$ as constants.

# 4 Exact Solution for An Arbitrary-Depth Linear Net

This section extends our result to multiple layers. We first derive the analytical formula for the global minimum of a general arbitrary-depth model. We then show that the landscape for a deeper network is highly nontrivial.

## 4.1 General Solution

The loss function is

$$\mathbb{E}_x \mathbb{E}_{\epsilon^{(1)},\epsilon^{(2)},...,\epsilon^{(D)}} \left( \sum_{i,i_1,i_2,...,i_D}^{d,d_1,d_2,...d_D} U_{i_D} \epsilon_{i_D}^{(D)} ... \epsilon_{i_2}^{(2)} W_{i_2 i_1}^{(2)} \epsilon_{i_1}^{(1)} W_{i_1 i}^{(1)} x_i - y \right)^2 + \gamma_u \|U\|^2 + \sum_{i=1}^{D} \gamma_i \|W^{(i)}\|^2, \tag{9}$$

where all the noises $\epsilon$ are independent, and for all $i$ and $j$, $\mathbb{E}[\epsilon_j^{(i)}] = 1$ and $\mathbb{E}[(\epsilon_j^{(i)})^2] = \sigma_i^2 + 1 > 1$. We first show that for general $D$, the global minimum exists for this objective.

**Proposition 1.** *For $D \geq 1$ and strictly positive $\gamma_u$, $\gamma_1, ..., \gamma_D$, the global minimum for Eq.(9) exists.*

Note that the positivity of the regularization strength is crucial. If one of the $\gamma_i$ is zero, the global minimum may not exist. The following theorem is our second main result.

**Theorem 2.** *Any global minimum of Eq. (9) is of the form*

$$\begin{cases} U = b_u \mathbf{r}_D; \\ W^{(i)} = b_i \mathbf{r}_i \mathbf{r}_{i-1}^T; \\ W^{(1)} = \mathbf{r}_1 \mathbb{E}[xy]^T (b_u \prod_{i=2}^D b_i) \mu \left[ (b_u \prod_{i=2}^D b_i)^2 s^2 (\sigma^2 + d_1) A_0 + \gamma_w I \right]^{-1}, \end{cases} \tag{10}$$

*where $\mu = \prod_{i=2}^D d_i$, $s^2 = \prod_{i=2}^D d_i(\sigma^2 + d_i)$, $b_u \geq 0$ and $b_i \geq 0$, and $\mathbf{r}_i = (\pm 1, ..., \pm 1)$ is an arbitrary vertex of a $d_i$-dimensional hypercube for all $i$. Furthermore, let $b_1 := \sqrt{\|W_{i:}\|^2/d}$ and $b_{D+1} := b_u$, $b_i$ satisfies*

$$\gamma_{k+1} d_{k+1} b_{k+1}^2 = \gamma_k d_{k-1} b_k^2. \tag{11}$$

*Proof Sketch.* We prove by induction on the depth $D$. The base case is proved in Theorem 1. We then show that for a general depth, the objective involves optimizing subproblems, one of which is a $D - 1$ layer problem that follows by the induction assumption, and the other is a two-layer problem that has been solved in Theorem 1. Putting these two subproblems together, one obtains Eq. (10). □

**Remark.** *We deal with the technical case of having a bias term for each layer in Appendix D. For example, we will show that if one has preprocessed the data such that $\mathbb{E}[x] = 0$ and $\mathbb{E}[y] = 0$, our main results remain precisely unchanged.*

The condition in Eq. (11) shows that the scaling factor $b_i$ for all $i$ is not independent of one another. This automatic balancing of the norm of all layers is a consequence of the rescaling invariance of the multilayer architecture and the use of weight decay. It is well-known that this rescaling invariance also exists in a neural network with the ReLU activation, and so this balancing condition is also directly relevant for ReLU networks.

Condition (11) implies that all the $b_i$ can be written in terms of one of the $b_i$:

$$b_u \prod_{i=2}^D b_i = c_0 \text{sgn}\left(b_u \prod_{i=2}^D b_i\right) |b_2^D| := c_0 \text{sgn}\left(b_u \prod_{i=2}^D b_i\right) b^D \tag{12}$$

where $c_0 = \frac{(\gamma_2 d_2 d_1)^{D/2}}{\sqrt{\gamma_u \prod_{i=2}^D \gamma_i} \prod_{i=2}^D d_i \sqrt{d_1}}$ and $b \geq 0$. Consider the first layer ($i = 1$), Eq (11) shows that the global minimum must satisfy the following equation, which is equivalent to a high-order polynomial in $b$ that does not have an analytical solution in general:

$$\|\mathbb{E}[xy]^T c_0 b^D \mu \left[ c_0^2 b^{2D} s^2 (\sigma^2 + d_1) A_0 + \gamma_w I \right]^{-1}\|^2 = d_2 b^2. \tag{13}$$

Thus, this condition is an extension of the condition (6) for two-layer networks.

At this point, it pays to clearly define the word "solution," especially given that it has a special meaning in this work because it now becomes highly nontrivial to differentiate between the two types of solutions.

**Definition 1.** *We say that a non-negative real $b$ is a solution if it satisfies Eq.* (13). *A solution is trivial if $b = 0$ and nontrivial otherwise.*

Namely, a global minimum must be a solution, but a solution is not necessarily a global minimum. We have seen that even in the two-layer case, the global minimum can be the trivial one when the strength of the signal is too weak or when the strength of regularization is too strong. It is thus natural to expect $0$ to be the global minimum under a similar condition, and one is interested in whether the condition becomes stronger or weaker as the depth of the model is increased. However, it turns out this naive expectation is not true. In fact, when the depth of the model is larger than $2$, the condition for the trivial global minimum becomes highly nontrivial.

The following proposition shows why the problem becomes more complicated. In particular, we have seen that in the case of a two-layer net, some elementary argument has helped us show that the trivial solution $b = 0$ is either a saddle or the global minimum. However, the proposition below shows that with $D \geq 2$, the landscape becomes more complicated in the sense that the trivial solution is always a local minimum, and it becomes difficult to compare the loss value of the trivial solution with the nontrivial solution because the value of $b^*$ is unknown in general.

**Proposition 2.** *Let $D \geq 2$ in Eq.* (9). *Then, the solution $U = 0$, $W^{(D)} = 0$, ..., $W^{(1)} = 0$ is a local minimum with a diagonal positive-definite Hessian $\gamma I$.*

Comparing the Hessian of $D \geq 2$ and $D = 1$, one notices a qualitative difference: for $D \geq 2$, the Hessian is always diagonal (at 0); for $D = 1$, in sharp contrast, the off-diagonal terms are nonzero in general, and it is these off-diagonal terms that can break the positive-definiteness of the Hessian. This offers a different perspective on why there is a qualitative difference between $D = 1$ and $D = 2$.

Lastly, note that, unlike the depth-1 case, one can no longer find a precise condition such that a $b \neq 0$ solution exists for a general $A_0$. The reason is that the condition for the existence of the solution is now a high-order polynomial with quite arbitrary intermediate terms. The following proposition gives a sufficient but stronger-than-necessary condition for the existence of a nontrivial solution, when all the $\sigma_i$, intermediate width $d_i$ and regularization strength $\gamma_i$ are the same.[4]

**Proposition 3.** *Let $\sigma_i^2 = \sigma^2 > 0$, $d_i = d_0$ and $\gamma_i = \gamma > 0$ for all $i$. Assuming $a_{\min} > 0$, the only solution is trivial if*

$$\frac{D+1}{2D}\|\mathbb{E}[xy]\|d_0^{D-1}\left(\frac{(D-1)\|\mathbb{E}[xy]\|}{2Dd_0(\sigma^2+d_0)^D a_{\min}}\right)^{\frac{D-1}{D+1}} < \gamma. \tag{14}$$

*Nontrivial solutions exist if*

$$\frac{D+1}{2D}\|\mathbb{E}[xy]\|d_0^{D-1}\left(\frac{(D-1)\|\mathbb{E}[xy]\|}{2Dd_0(\sigma^2+d_0)^D a_{\max}}\right)^{\frac{D-1}{D+1}} \geq \gamma. \tag{15}$$

*Moreover, the nontrivial solutions are both lower and upper-bounded:[5]*

$$\frac{1}{d_0}\left[\frac{\gamma}{\|\mathbb{E}[xy]\|}\right]^{\frac{1}{D-1}} \leq b^* \leq \left[\frac{\|\mathbb{E}[xy]\|}{d_0(\sigma^2+d_0)^D a_{\max}}\right]^{\frac{1}{D+1}}. \tag{16}$$

*Proof Sketch.* The proof follows from the observation that the l.h.s. of Eq. (13) is a continuous function and must cross the r.h.s. under certain sufficient conditions. □

One should compare the general condition here with the special condition for $D = 1$. One sees that for $D \geq 2$, many other factors (such as the width, the depth, and the spectrum of the data covariance $A_0$) come into play to determine the existence of a solution apart from the signal strength $\mathbb{E}[xy]$ and the regularization strength $\gamma$.

---

[4]This is equivalent to setting $c_0 = \sqrt{d_0}$. The result is qualitatively similar but involves additional factors of $c_0$ if $\sigma_i$, $d_i$, and $\gamma_i$ all take different values. We thus only present the case when $\sigma_i$, $d_i$, and $\gamma_i$ are the same for notational concision and for emphasizing the most relevant terms. Also, note that this proposition gives a *sufficient and necessary* condition if $A_0 = \sigma_x^2 I$ is proportional to the identity.

[5]For $D = 1$, we define the lower-bound as $\lim_{\eta \to 0^+} \lim_{D \to 1^+} \frac{1}{d_0}\left[\frac{\gamma+\eta}{\|\mathbb{E}[xy]\|}\right]^{\frac{1}{D-1}}$, which equal to zero if $\mathbb{E}[xy] \geq \gamma$, and $\infty$ if $\mathbb{E}[xy] < \gamma$. With this definition, this proposition applies to a two-layer net as well.

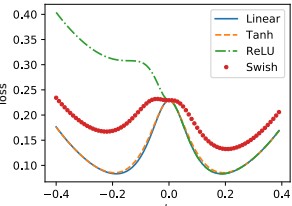 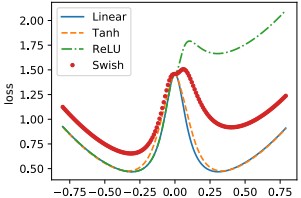

Figure 2: The training loss as a function of $b$ for a $D = 1$ network with different activation functions in the hidden layer. For simplicity, dropout is not implemented. The non-linear activation functions we considered are ReLU, Tanh, and Swish. The left and right panels use different data. **Left:** $X$ are Gaussian random vectors, and $y = v \cdot x$ is a linear function of $x$. **Right:** $x$ are Gaussian random vectors, and $y = v \cdot tanh(x)$ are nonlinear functions of data; the weight $v$ is obtained as a Gaussian random vector.

### 4.2 Which Solution is the Global Minimum?

Again, we set $\gamma_i = \gamma > 0$, $\sigma_i^2 = \sigma^2 > 0$ and $d_i = d_0 > 0$ for all $i$ for notational concision. Using this condition and applying Lemma 3 to Theorem 2, the solution now takes the following form, where $b \geq 0$,

$$
\begin{cases}
U = \sqrt{d_0} b \mathbf{r}_D; \\
W^{(i)} = b \mathbf{r}_i \mathbf{r}_{i-1}^T; \\
W^{(1)} = \mathbf{r}_1 \mathbb{E}[xy]^T d_0^{D-\frac{1}{2}} b^D \left[ d_0^D (\sigma^2 + d_0)^D b^{2D} A_0 + \gamma \right]^{-1}.
\end{cases}
\tag{17}
$$

The following theorem gives a sufficient condition for the global minimum to be nontrivial. It also shows that the landscape of the linear net becomes complicated and can contain more than 1 local minimum.

**Theorem 3.** *Let $\sigma_i^2 = \sigma^2 > 0$, $d_i = d_0$ and $\gamma_i = \gamma > 0$ for all $i$ and assuming $a_{\min} > 0$. Then, if*

$$
\|\mathbb{E}[xy]\|^2 \geq \frac{\gamma^{\frac{D+1}{D}} D^2 (\sigma^2 + d_0)^{D-1} a_{max}^{\frac{D-1}{D}}}{d_0^{D-1} (D-1)^{\frac{D-1}{D}}}
\tag{18}
$$

*the global minimum of Eq. (9) is one of the nontrivial solutions.*

While there are various ways this bound can be improved, it is general enough for our purpose. In particular, one sees that, for a general depth, the condition for having a nontrivial global minimum depends not only on the $\mathbb{E}[xy]$ and $\gamma$ but also on the model architecture in general. For a more general architecture with different widths etc., the architectural constant $c_0$ from Eq. (13) will also enter the equation. In the limit of $D \rightarrow 1^+$, relation (18) reduces to

$$
\|\mathbb{E}[xy]\|^2 \geq \gamma^2,
\tag{19}
$$

which is the condition derived for the 2-layer case.

## 5 Implications

**Relevance to nonlinear models.** We first caution the readers that the following discussion should be taken with a caveat and is based on the philosophy that deep linear nets can approximate the nonlinear ones. This approximation certainly holds for fully connected models with differentiable activation functions such as tanh or Swish because they are, up to first-order Taylor expansion, a deep linear net around zero, which is the region for which our theory is the most relevant. We empirically demonstrate that close to the origin, the landscape of linear nets can indeed approximate that of nonlinear nets quite well. To compare, we plug in the solution in Theorem 4 to both linear and nonlinear models of the same architecture and compare the loss values at different values of $b$ around $b = 0$. For simplicity, we only consider the case $D = 1$. The activation functions we consider are ReLU, Tanh, and Swish (Ramachandran et al., 2017), a modern and differentiable variant of ReLU. See Fig. 2.

The regressor $x \in \mathbb{R}^d$ is sampled as Gaussian random vectors. We consider two methods of generating $y$; the first one (left) is $y = v \cdot x$. The second one (right) is $y = v \cdot \tanh(x)$, where the weight $v \in \mathbb{R}^d$ is obtained as a Gaussian random vector. Fig. 2 shows that the landscape consisting of Tanh is always close to the linear landscape. Swish is not as good as Tanh, but the Swish landscape shows a similar tendency to the linear landscape. The ReLU landscape is not so close to the linear landscape

either for $b > 0$ or $b < 0$, but it agrees completely with the linear landscape on the other side, as expected. Besides the quantitative closeness, it is also important to note that all the landscapes agree qualitatively, containing the same number of local minima at similar values of $b$.

**Landscape of multi-layer neural networks**. The combination of Theorem 3 and Proposition 2 shows that the landscape of a deep neural network can become highly nontrivial when there is a weight decay and when the depth of the model is larger than 2. This gives an incomplete but meaningful picture of a network's complicated but interesting landscape beyond two layers (see Figure 1 for an incomplete summary of our results). In particular, even when the nontrivial solution is the global minimum, the trivial solution is still a local minimum that needs to be escaped. Our result suggests the previous understanding that all local minima of a deep linear net are global cannot generalize to many practical settings where deep learning is found to work well. For example, a series of works attribute the existence of bad (non-global) minima to the use of nonlinearities (Kawaguchi, 2016) or the use of a non-regular (non-differentiable) loss function (Laurent and Brecht, 2018). Our result, in contrast, shows that the use of a simple weight decay is sufficient to create a bad minimum.[6] Moreover, the problem with such a minimum is two-fold: (1) (optimization) it is not global and so needs to be "overcome" and (2) (generalization) it is a minimum that has not learned any feature at all because the model constantly outputs zero. To the best of our knowledge, previous to our work, there has not been any proof that a bad minimum can generically exist in a rather arbitrary network without any restriction on the data.[7] Thus, our result offers direct and solid theoretical justification for the widely believed importance of escaping local minima in the field of deep learning (Kleinberg et al., 2018; Liu et al., 2021; Mori et al., 2022). In particular, previous works on escaping local minima often hypothesize landscapes that are of unknown relevance to an actual neural network. With our result, this line of research can now be established with respect to landscapes that are actually deep-learning-relevant.

Previous works also argue that having a deeper depth does not create a bad minimum (Lu and Kawaguchi, 2017). While this remains true, its generality and applicability to practical settings now also seem low. Our result shows that as long as weight decay is used, and as long as $D \geq 2$, there is indeed a bad local minimum at $0$. In contrast, there is no bad minimum at $0$ for a depth-2 network: the point $b = 0$ is either a saddle or the global minimum.[8] Having a deeper depth thus alters the qualitative nature of the landscape, and our results agree better with the common observation that a deeper network is harder, if not impossible, to optimize.

We note that our result can also be relevant for more modern architectures such as the ResNet. Using ResNet, one needs to change the dimension of the hidden layer after every bottleneck, and a learnable linear transformation is applied here. Thus, the "effective depth" of a ResNet would be roughly between the number of its bottlenecks and its total number of blocks. For example, a ResNet18 applied to CIFAR10 often has five bottlenecks and 18 layers in total. We thus expect it to have qualitatively similar behavior to a deep linear net with a depth in between. See Figure 1. The experimental details are given in Section A.

**Learnability of a neural network**. Now we analyze the solution when $D$ tends to infinity. We first note that the existence condition bound in (15) becomes exponentially harder to satisfy as $D$ becomes large:

$$\|\mathbb{E}[xy]\|^2 \geq 4d_0^2 a_{\max} \gamma e^{D \log[(\sigma^2 + d_0)/d_0]} + O(1). \tag{20}$$

When this bound is not satisfied, the given neural network cannot learn the data. Recall that for a two-layer net, the existence condition is nothing but $\|\mathbb{E}[xy]\|^2 > \gamma^2$, independent of the depth, width, or stochasticity in the model. For a deeper network, however, every factor comes into play, and the

---

[6]Some previous works do suggest the existence of bad minima when weight decay is present, but no direct proof exists yet. For example, Taghvaei et al. (2017) shows that when the model is approximated by a linear dynamical system, regularization can cause bad local minima. Mehta et al. (2021) shows the existence of bad local minima in deep linear networks with weight decay through numerical simulations.

[7]In the case of nonlinear networks without regularization, a few works proved the existence of bad minima. However, the previous results strongly depend on the data and are rather independent of architecture. For example, one major assumption is that the data cannot be perfected and fitted by a linear model (Yun et al., 2018; Liu, 2021; He et al., 2020). Some other works explicitly construct data distribution (Safran and Shamir, 2018; Venturi et al., 2019). Our result, in contrast, is independent of the data.

[8]Of course, in practice, the model trained with SGD can still converge to the trivial solution even if it is a saddle point (Ziyin et al., 2021) because minibatch SGD is, in general, not a good estimator of the local minima.

architecture of the model has a strong (and dominant) influence on the condition. In particular, a factor that increases polynomially in the model width and exponentially in the model depth appears.

A practical implication is that the use of weight decay may be too strong for deep networks. If one increases the depth or width of the model, one should also roughly decrease $\gamma$ according to Eq. (20).

**Insufficiency of the existing initialization schemes**. We have shown that $0$ is often a bad local minimum for deep learning. Our result further implies that escaping this local minimum can be highly practically relevant because standard initialization schemes are trapped in this local minimum for tasks where the signal $\mathbb{E}[xy]$ is weak. See Inequality (16): any nontrivial global minimum is lower-bounded by a factor proportional to $(\gamma/\|\mathbb{E}[xy]\|^{1/(D-1)})/d_0$, which can be seen as an approximation of the radius of the local minimum at the origin. In comparison, standard deep learning initialization schemes such as Kaiming init. initialize at a radius roughly $1/\sqrt{d_0}$. Thus, for tasks $\mathbb{E}[xy] \ll \gamma/\sqrt{d_0}$, these initialization methods are likely to initialize the model in the basin of attraction of the trivial regime, which can cause a serious failure in learning. To demonstrate, we perform a numerical simulation shown in the right panel of Figure 3, where we train $D = 2$ nonlinear networks with width 32 with SGD on tasks with varying $\|\mathbb{E}[xy]\|$. For sufficiently small $\|\mathbb{E}[xy]\|$, the model clearly is stuck at the origin.[9] In contrast, linear regression is never stuck at the origin. Our result thus suggests that it may be desirable to devise initialization methods that are functions of the data distribution.

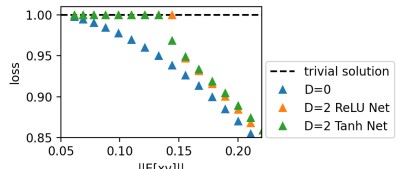

Figure 3: Training loss of $D = 2$ neural networks with ReLU and Tanh activations across synthetic tasks with different $\|\mathbb{E}[xy]\|$. We see that with the Kaiming initialization, both the Tanh net and the ReLU net are stuck at the trivial solution in expectation of our theory. In contrast, an optimized linear regressor ($D = 0$) is better than the trivial solution when $\|\mathbb{E}[xy]\| > 0$. See Section A for experimental details.

**Prediction variance of stochastic nets**. A major extension of the standard neural networks is to make them stochastic, namely, to make the output a random function of the input. In a broad sense, stochastic neural networks include neural networks trained with dropout (Srivastava et al., 2014; Gal and Ghahramani, 2016), Bayesian networks (Mackay, 1992), variational autoencoders (VAE) (Kingma and Welling, 2013), and generative adversarial networks (Goodfellow et al., 2014). Stochastic networks are thus of both practical and theoretical importance to study. Our result can also be used for studying the theoretical properties of stochastic neural networks. Here, we present a simple application of our general solution to analyze the properties of a stochastic net. The following theorem summarizes our technical results.

**Theorem 4.** *Let* $\sigma_i^2 = \sigma^2 > 0$, $d_i = d_0$ *and* $\gamma_i = \gamma > 0$ *for all* $i$. *Let* $A_0 = \sigma_x^2 I$. *Then, at any global minimum of Eq.* (9)*, holding other parameters fixed,*

1. *in the limit of large* $d_0$, $Var[f(x)] = O\left(d_0^{-1}\right)$;
2. *in the limit of large* $\sigma^2$, $Var[f(x)] = O\left(\frac{1}{(\sigma^2)^D}\right)$;
3. *In the limit of large* $D$, $Var[f(x)] = O\left(e^{-2D \log[(\sigma^2+d_0)/d_0]}\right)$.

Interestingly, the scaling of prediction variance in asymptotic $\sigma^2$ is different for different widths. The third result shows that the prediction variance decreases exponentially fast in $D$. In particular, this result answers a question recently proposed in Ziyin et al. (2022b): does a stochastic net trained on MSE have a prediction variance that scales towards $0$? We improve on their result in the case of a deep linear net by (a) showing that the $d_0^{-1}$ is tight in general, independent of the depth or other factors of the model, and (b) proving a bound showing that the variance also scales towards zero as depth increases, which is a novel result of our work. Our result also offers an important insight into the cause of the vanishing prediction variance. Previous works (Alemi et al., 2018) often attribute the cause to the fact that a wide neural network is too expressive. However, our result implies that this is not always the case because a linear network with limited expressivity can also have a vanishing variance as the model tends to an infinite width.

**Collapses in deep learning.** Lastly, we comment briefly on the apparent similarity between different types of collapses that occur in deep learning. For neural collapse, our result agrees with the recent

---

[9]There are many natural problems where the signal is extremely weak. One well-known example is the problem of future price prediction in finance, where the fundamental theorem of finance forbids a large $\|\mathbb{E}[xy]\|$ (Fama, 1970).

works that identify weight decay as a main cause (Rangamani and Banburski-Fahey, 2022). For Bayesian deep learning, Wang and Ziyin (2022) identified the cause of the posterior collapse in a two-layer VAE structure to be that the regularization of the mean of the latent variable $z$ is too strong. More recently, the origin and its stability have also been discussed as the dimensional collapse in self-supervised learning (Ziyin et al., 2022a). Although appearing in different contexts of deep learning, the three types of collapses share the same phenomenology that the model converges to a "collapsed" regime where the learned representation becomes low-rank or constant, which agrees with the behavior of the trivial regime we identified. We refer the readers to Ziyin and Ueda (2022) for a study of how the second-order phase transition framework of statistical physics can offer a possible unified explanation of these phenomena.

## 6 Conclusion

In this work, we derived the exact solution of a deep linear net with arbitrary depth and width and with stochasticity. Our work sheds light on the highly complicated landscape of a deep neural network. Compared to the previous works that mostly focus on the qualitative understanding of the linear net, our result offers a more precise quantitative understanding of deep linear nets. Quantitative understanding is one major benefit of knowing the exact solution, whose usefulness we have also demonstrated with the various implications. The results, although derived for linear models, are also empirically shown to be relevant for networks with nonlinear activations. Lastly, our results strengthen the line of thought that analytical approaches to deep linear models can be used to understand deep neural networks, and it is the sincere hope of the authors to attract more attention to this promising field.

## Acknowledgement

Ziyin is financially supported by the GSS scholarship of the University of Tokyo and the JSPS fellowship. Li is financially supported by CNRS. X. Meng is supported by JST CREST Grant Number JPMJCR1912, Japan.

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
