# A  Experimental Details

For the experiment in Figure 3, the input data consists of 1000 data points sampled from a multivariate Gaussian distribution: $x \sim \mathcal{N}(0, I_5)$. The target is generated by a linear transformation $y = v \cdot x$, where the norm of $v$ is rescaled to obtain different values of $\|\mathbb{E}[xy]\|$ as the control parameter of the simulation. The models are with $D = 2$ neural networks with bias terms and with hidden width 32 for both hidden layers. The training proceeds with gradient descent with a learning rate of 0.1 for $10^4$ iterations when the training loss has stopped decreasing for all the experiments.

For the CIFAR10 experiments, we train a standard ResNet18 with roughly $10^7$ parameters under the standard procedure, with a batch size of 256 for 100 epochs.[10] For the linear models, we use a hidden width of 32 without any bias term. The training proceeds with SGD with batch size 256 for 100 epochs with a momentum of 0.9. The learning rate is 0.002, chosen as the best learning rate from a grid search over $[0.001, 0.002, ..., 0.01]$.

# B  Proofs

## B.1  Proof of Lemma 1

*Proof.* Note that the first term in the loss function is invariant to the following rescaling for any $a > 0$:

$$\begin{cases} U_i \to aU_i; \\ W_{ij} \to W_{ij}/a; \end{cases} \tag{21}$$

meanwhile, the $L_2$ regularization term changes as $a$ changes. Therefore, the global minimum must have a minimized $a$ with respect to any $U$ and $W$.

One can easily find the solution:

$$a^* = \arg\min_a \left( \gamma_u a^2 U_i^2 + \gamma_w \sum_j \frac{W_{ij}^2}{a^2} \right) = \left( \frac{\gamma_w \sum_j W_{ij}^2}{\gamma_u U_i^2} \right)^{1/4}. \tag{22}$$

Therefore, at the global minimum, we must have $\gamma_u a^2 U_i^2 = \gamma_w \sum_j \frac{W_{ij}^2}{a^2}$, so that

$$(U_i^*)^2 = (a^* U_i)^2 = \frac{\gamma_w}{\gamma_u} \sum_j (W_{ij}^*)^2, \tag{23}$$

which completes the proof. □

## B.2  Proof of Lemma 2

*Proof.* By Lemma 1, we can write $U_i$ as $b_i$ and $W_{i:}$ as $b_i w_i$ where $w_i$ is a unit vector, and finding the global minimizer of Eq. (2) is equivalent to finding the minimizer of the following objective,

$$\mathbb{E}_{x,\varepsilon} \left[ \left( \sum_{i,j} b_i^2 \epsilon_i w_{ij} x_j - y \right)^2 \right] + (\gamma_u + \gamma_w) \|b\|_2^2, \tag{24}$$

$$= \mathbb{E}_x \left[ \left( \sum_{i,j} b_i^2 w_{ij} x_j - y \right)^2 \right] + \sigma^2 \sum_{ij} b_i^4 \left( \sum_k w_{ik} x_k \right)^2 + (\gamma_u + \gamma_w) \|b\|_2^2, \tag{25}$$

The lemma statement is equivalent to $b_i = b_j$ for all $i$ and $j$.

We prove this by contradiction. Suppose there exist $i$ and $j$ such that $b_i \neq b_j$, we can choose $i$ to be the index of $b_i$ with maximum $b_i^2$, and let $j$ be the index of $b_j$ with minimum $b_j^2$. Now, we can construct a different solution by the following replacement of $b_i w_{i:}$ and $b_j w_{j:}$:

$$\begin{cases} b_i^2 w_{i:} \to c^2 v; \\ b_j^2 w_{j:} \to c^2 v, \end{cases} \tag{26}$$

---

[10]Specifically, we use the implementation and training procedure of https://github.com/kuangliu/pytorch-cifar, with standard augmentations such as random crop, etc.

where $c$ is a positive scalar and $v$ is a unit vector such that $2c^2 v = b_i^2 w_{i:} + b_j^2 w_{j:}$. Note that, by the triangular inequality, $2c^2 \leq b_i^2 + b_j^2$. Meanwhile, all the other terms, $b_k$ for $k \neq i$ and $k \neq j$, are left unchanged. This transformation leaves the first term in the loss function (25) unchanged, and we now show that it decreases the other terms.

The change in the second term is

$$\left(b_i^2 \sum_k w_{ik} x_k\right)^2 + \left(b_j^2 \sum_k w_{jk} x_k\right)^2 \to 2\left(c^2 \sum_k v_k x_k\right)^2 = \frac{1}{2}\left(b_i^2 \sum_k w_{ik} x_k + b_j^2 \sum_k w_{jk} x_k\right)^2. \quad (27)$$

By the inequality $a^2 + b^2 \geq (a + b)^2/2$, we see that the left-hand side the larger than the right-hand side.

We now consider the $L_2$ regularization term. The change is

$$(\gamma_u + \gamma_w)(b_i^2 + b_j^2) \to 2(\gamma_u + \gamma_w)c^2, \quad (28)$$

and the left-hand side is again larger than the right-hand side by the inequality mentioned above: $2c^2 \leq b_i^2 + b_j^2$. Therefore, we have constructed a solution whose loss is strictly smaller than that of the global minimum: a contradiction. Thus, the global minimum must satisfy

$$U_i^2 = U_j^2 \quad (29)$$

for all $i$ and $j$.

Likewise, we can show that $U_i W_{i:} = U_j W_{j:}$ for all $i$ and $j$. This is because the triangular inequality $2c^2 \leq b_i^2 + b_j^2$ is only an equality if $U_i W_{i:} = U_j W_{j:}$. If $U_i W_{i:} \neq U_j W_{j:}$, following the same argument above, we arrive at another contradiction. □

## B.3  Proof of Theorem 1

*Proof.* By Lemma 2, at any global minimum, we can write $U_* = b\mathbf{r}$ for some $b \in \mathbb{R}$. We can also write $W_* = \mathbf{r}v^T$ for a general vector $v \in \mathbb{R}^d$. Without loss of generality, we assume that $b > 0$ (because the sign of $b$ can be absorbed into $\mathbf{r}$).

The original problem in Eq. (2) is now equivalently reduced following problem because $\mathbf{r}^T\mathbf{r} = d_1$:

$$\min_{b,v} \mathbb{E}_x\left[\left(bd_1 \sum_j v_j x_j - y\right)^2 + b^2 d_1 \sigma^2 \left(\sum_k v_k x_k\right)^2\right] + \gamma_u d_1 b^2 + \gamma_w d_1 \|v\|_2^2. \quad (30)$$

For any fixed $b$, the global minimum of $v$ is well known:[11]

$$v = b\mathbb{E}[xy]^T \left[b^2 \left(\sigma^2 + d_1\right) A_0 + \gamma_w I\right]^{-1}. \quad (31)$$

By Lemma 1, at a global minimum, $b$ also satisfies the following condition:

$$b^2 = \frac{\gamma_w}{\gamma_u}\|v\|^2, \quad (32)$$

One solution to this equation is $b = 0$, and we are interested in whether solutions with $b \neq 0$ exist. If there is no other solution, then $b = 0$ must be the unique global minimum; otherwise, we need to identify which of the solutions are actual global minima. When $b \neq 0$,

$$\left\|\left[b^2 \left(\sigma^2 + d_1\right) A_0 + \gamma_w I\right]^{-1} \mathbb{E}[xy]\right\|^2 = \frac{\gamma_u}{\gamma_w}. \quad (33)$$

Note that the left-hand side is monotonically decreasing in $b^2$, and is equal to $\gamma_w^{-2}\|\mathbb{E}[xy]\|^2$ when $b = 0$. When $b \to \infty$, the left-hand side tends to 0. Because the left-hand side is a continuous and monotonic function of $b$, a unique solution $b_* > 0$ that satisfies Eq. (33) exists if and only if $\gamma_w^{-2}\|\mathbb{E}[xy]\|^2 > \gamma_u/\gamma_w$, or,

$$\|\mathbb{E}[xy]\|^2 > \gamma_u \gamma_w. \quad (34)$$

---

[11]Namely, it is the solution of a ridgeless linear regression problem.

Therefore, at most, three candidates for global minima of the loss function exist:

$$\begin{cases} b = 0, \; v = 0 & \text{if } \|\mathbb{E}[xy]\|^2 \le \gamma_u \gamma_w; \\ b = \pm b_*, \; v = b\big[b^2\left(\sigma^2 + d_1\right) A_0 + \gamma_w I\big]^{-1} \mathbb{E}[xy], & \text{if } \|\mathbb{E}[xy]\|^2 > \gamma_u \gamma_w, \end{cases} \tag{35}$$

where $b^* > 0$.

In the second case, one needs to discern the saddle points from the global minima. Using the expression of $v$, one finds the expression of the loss function as a function of $b$

$$d_1(d_1 + \sigma^2)b^4 \sum_i \frac{\mathbb{E}[x'y]_i^2 a_i}{[b^2(\sigma^2 + d_1)a_i + \gamma_w]^2} - 2b^2 d_1 \sum_i \frac{\mathbb{E}[x'y]_i^2}{b^2(\sigma^2 + d_1)a_i + \gamma_w} + \mathbb{E}[y^2]$$
$$+ \gamma_u d_1 b^2 + \gamma_w d_1 \sum_i \frac{\mathbb{E}[x'y]_i^2 b^2}{[b^2(\sigma^2 + d_1)a_i + \gamma_w]^2}, \tag{36}$$

where $x' = Rx$ such that $RA_0 R^{-1}$ is a diagonal matrix. We now show that condition (34) is sufficient to guarantee that $0$ is not the global minimum.

At $b = 0$, the first nonvanishing derivative of $b$ is the second-order derivative. The second order derivative at $b = 0$ is

$$-2d_1\|\mathbb{E}[xy]\|^2/\gamma_w + 2\gamma_u d_1, \tag{37}$$

which is negative if and only if $\|\mathbb{E}[xy]\|^2 > \gamma_u \gamma_w$. If the second derivative at $b = 0$ is negative, $b = 0$ cannot be a minimum. It then follows that for $\|\mathbb{E}[xy]\|^2 > \gamma_u \gamma_w$, $b = \pm b^*$, $v = b\big[b^2\left(\sigma^2 + d_1\right) A_0 + \gamma_w I\big]^{-1} \mathbb{E}[xy]$, if $\|\mathbb{E}[xy]\|^2 > \gamma_u \gamma_w$ are the two global minimum (because the loss is invariant to the sign flip of $b$). For the same reason, when $\|\mathbb{E}[xy]\|^2 < \gamma_u \gamma_w$, $b = 0$ gives the unique global minimum. This finishes the proof. □

### B.4 Proof of Proposition 1

*Proof.* We first show that there exists a constant $r$ such that the global minimum must be confined within a (closed) $r$-Ball around the origin. The objective (9) can be upper-bounded by

$$Eq. (9) \ge \gamma_u \|U\|^2 + \sum_{i=1}^{D} \gamma_i \|W^{(i)}\|^2 \ge \gamma_{\min}\left(\|U\|^2 + \sum_i \|W^{(i)}\|^2\right), \tag{38}$$

where $\gamma_{\min} := \min_{i \in \{u,1,2,...,D\}} > 0$. Now, let $w$ denote be the union of all the parameters $(U, W^{(i)})$ and viewed as a vector. We see that the above inequality is equivalent to

$$Eq. (9) \ge \gamma_{\min}\|w\|^2. \tag{39}$$

Now, note that the loss value at the origin is $\mathbb{E}[y^2]$, which means that for any $w$, whose norm $\|w\|^2 \ge \mathbb{E}[y^2]/\gamma_{\min}$, the loss value must be larger than the loss value of the origin. Therefore, let $r = \mathbb{E}[y^2]/\gamma_{\min}$, we have proved that the global minimum must lie in a closed $r$-Ball around the origin.

As the last step, because the objective is a continuous function of $w$ and the $r$-Ball is a compact set, the minimum of the objective in this $r$-Ball is achievable. This completes the proof. □

### B.5 Proof of Theorem 2

We divide the proof into the proof of a proposition and a lemma, and combining the following proposition and lemma obtains the theorem statement.

#### B.5.1 Proposition 4

**Proposition 4.** *Any global minimum of Eq. (9) is of the form*

$$\begin{cases} U = b_u \mathbf{r}_D; \\ W^{(i)} = b_i \mathbf{r}_i \mathbf{r}_{i-1}^T; \\ W^{(1)} = \mathbf{r}_1 \mathbb{E}[xy]^T (b_u \prod_{i=2}^{D} b_i)\mu \big[(b_u \prod_{i=2}^{D} b_i)^2 s^2 \left(\sigma^2 + d_1\right) A_0 + \gamma_w I\big]^{-1}, \end{cases} \tag{40}$$

*where $\mu = \prod_{i=2}^{D} d_i$, $s^2 = \prod_{i=2}^{D} d_i(\sigma^2 + d_i)$, $b_u \ge 0$ and $b_i \ge 0$, and $\mathbf{r}_i = (\pm 1, ..., \pm 1)$ is an arbitrary vertex of a $d_i$-dimensional hypercube for all $i$.*

*Proof.* Note that the trivial solution is also a special case of this solution with $b = 0$. We thus focus on deriving the form of the nontrivial solution.

We prove by induction on $D$. The base case with depth 1 is proved in Theorem 1. We now assume that the same holds for depth $D - 1$ and prove that it also holds for depth $D$.

For any fixed $W^{(1)}$, the loss function can be equivalently written as

$$\mathbb{E}_{\tilde{x}}\mathbb{E}_{\epsilon^{(2)},...,\epsilon^{(D)}}\left(\sum_{i_1,i_2,...,i_D}^{d_1,d_2,...d_D} U_{i_D}\epsilon_{i_D}^{(D)}...\epsilon_{i_2}^{(2)}W_{i_2i_1}^{(2)}\tilde{x}_{i_1} - y\right)^2 + \gamma_u\|U\|^2 + \sum_{i=2}^{D}\gamma_i\|W^{(i)}\|^2 + const., \tag{41}$$

where $\tilde{x} = \epsilon_{i_1}^{(1)}\sum_i W_{i_1i}^{(1)}x_i$. Namely, we have reduced the problem to a problem involving only a depth $D - 1$ linear net with a transformed input $\tilde{x}$.

By the induction assumption, the global minimum of this problem takes the form of Eq. (10), which means that the loss function can be written in the following form:

$$\mathbb{E}_{\tilde{x}}\mathbb{E}_{\epsilon^{(2)},...,\epsilon^{(D)}}\left(b_ub_D...b_3\sum_{i_1,i_2,...,i_D}^{d_1,d_2,...d_D}\epsilon_{i_D}^{(D)}...\epsilon_{i_2}^{(2)}v_{i_1}\tilde{x}_{i_1} - y\right)^2 + L_2 \ reg., \tag{42}$$

for an arbitrary optimizable vector $v_{i_1}$. The term $\sum_{i_2,...,i_D}^{d_2,...d_D}\epsilon_{i_D}^{(D)}...\epsilon_{i_2}^{(2)} := \eta$ can now be regarded as a single random variable such that $\mathbb{E}[\eta] = \prod_{i=2}^{D}d_i := \mu$ and $\mathbb{E}[\eta^2] = \prod_{i=2}^{D}d_i(\sigma_i^2 + d_i) := s^2$. Computing the expectation over all the noises except for $\epsilon^{(1)}$, one finds

$$\mathbb{E}_{\tilde{x}}\left(b_ub_D...b_3s\sum_{i_1}v_{i_1}\tilde{x}_{i_1} - \frac{\mu y}{s}\right)^2 + L_2 \ reg. + const. \tag{43}$$

$$= \mathbb{E}_{x,\epsilon^{(1)}}\left(b_ub_D...b_3s\sum_{i,i_1}v_{i_1}\epsilon_{i_1}^{(1)}W_{i_1i}^{(1)}x_i - \frac{\mu y}{s}\right)^2 + L_2 \ reg. + const., \tag{44}$$

where we have ignored the constant term because it does not affect the minimizer of the loss. Namely, we have reduced the original problem to a two-layer linear net problem where the label becomes effectively rescaled for a deep network.

For any fixed $b_u, ..., b_3$, we can define $\bar{x} := b_ub_D...b_3sx$, and obtain the following problem, whose global minimum we have already derived:

$$\mathbb{E}_{\bar{x}}\mathbb{E}_{\epsilon^2,...,\epsilon_D}\left(\sum_{i,i_1}v_{i_1}W_{i_1i}^{(1)}\bar{x}_i - \frac{\mu y}{s}\right)^2. \tag{45}$$

By Theorem 1, the global minimum is identically 0 if $\|\mathbb{E}[\mu\bar{x}y/s]\|^2 < d_2\gamma_2\gamma_1$, or, $\mathbb{E}[xy] \leq \frac{\gamma_2\gamma_1}{b_3^2...b_u^2(\prod_{i=3}^{D}d_i)}$. When $\mathbb{E}[xy] > \frac{\gamma_2\gamma_1}{b_3^2...b_u^2(\prod_{i=3}^{D}d_i)}$, the solution can be non-trivial:

$$\begin{cases} v_* = b_2^*\mathbf{r}_1; \\ W_* = \mathbf{r}_1\mathbb{E}[xy]^T\mu b_2^*b_3...b_u\left[(b_2^*)^2d_3^2...d_D^2b_u^2s^2\left(\sigma^2 + d_1\right)A_0 + \gamma_1 I\right]^{-1}, \end{cases} \tag{46}$$

for some $b_2^*$. This proves the theorem. □

### B.6   Lemma 3

**Lemma 3.** *At any global minimum of Eq. (9), let $b_1 := \sqrt{\|W_{i:}\|^2/d}$ and $b_{D+1} := b_u$,*

$$\gamma_{k+1}d_{k+1}b_{k+1}^2 = \gamma_k d_{k-1}b_k^2. \tag{47}$$

*Proof.* It is sufficient to show that for all $k$ and $i$,

$$\gamma_{k+1}\sum_{ij}(W_{ji}^{k+1})^2 = \gamma_k\sum_{ij}(W_{ij}^k)^2. \tag{48}$$

We prove by contradiction. Let $U^*, W^*$ be the global minimum of the loss function. Assuming that for an arbitrary $k$,

$$\gamma_{k+1} \sum_{ij} (W_{ji}^{*,k+1})^2 \neq \gamma_k \sum_{ij} (W_{ij}^{*,k})^2. \tag{49}$$

Introduce $W^a$ such that $W_{ji}^{a,k+1} = aW_{ji}^{*,k+1}$ and $W_{ji}^{a,k} = W_{ji}^{*,k}/a$. The loss without regularization is invariant under the transformation of $W^* \to W^a$, namely

$$L_0(W^*) = L_0(W^a). \tag{50}$$

In the regularization, all the terms remain invariant except two terms:

$$\begin{cases} \gamma_{k+1} \sum_{ij} (W_{ji}^{*,k+1})^2 \to \gamma_{k+1} \sum_{ij} (W_{ji}^{a,k+1})^2 = a^2 \gamma_{k+1} \sum_{ij} (W_{ji}^{*,k+1})^2 \\ \gamma_k \sum_{ij} (W_{ij}^{*,k})^2 \to \gamma_k \sum_{ij} (W_{ji}^{a,k})^2 = a^{-2} \gamma_k \sum_{ij} (W_{ji}^{*,k})^2 \end{cases} \tag{51}$$

It could be shown that, the sum of $a^2 \gamma_{k+1} \sum_{ij} (W_{ji}^{*,k+1})^2$ and $a^{-2} \gamma_k \sum_{ij} (W_{ji}^{*,k})^2$ reaches its minimum when $a^2 = \sqrt{\frac{\gamma_k \sum_{ij} (W_{ji}^{*,k})^2}{\gamma_{k+1} \sum_{ij} (W_{ji}^{*,k+1})^2}}$. If $\gamma_{k+1} \sum_{ij} (W_{ji}^{*,k+1})^2 \neq \gamma_k \sum_{ij} (W_{ij}^{*,k})^2$, one can choose $a$ to minimize the regularization terms in the loss function such that $L(W^a) < L(W^*)$, indicating $W^*$ is not the global minimum. Thus, $\gamma_{k+1} \sum_{ij} (W_{ji}^{*,k+1})^2 \neq \gamma_k \sum_{ij} (W_{ij}^{*,k})^2$ cannot be true. $\square$

### B.7 Proof of Proposition 2

*Proof.* Let

$$L_0 = \mathbb{E}_{\tilde{x}} \mathbb{E}_{\epsilon^2, \ldots, \epsilon_D} \left( \sum_{i_1, i_2, \ldots, i_D}^{d_1, d_2, \ldots d_D} U_{i_D} \epsilon_{i_D}^{(D)} \ldots \epsilon_{i_1}^{(1)} W_{i_1 i}^{(1)} x_i - y \right)^2. \tag{52}$$

$L_0$ is a polynomial containing $2D + 2$th order, $D + 1$th order, and 0th order terms in terms of parameters $U$ and $W$. The second order derivative of $L$ is thus a polynomial containing $2D$-th order and $(D-1)$-th order terms; however, other orders are not possible. For $D \geq 2$, there are no constant terms in the Hessian of $L$, and there is at least a parameter in each of the terms.

The Hessian of the full loss function with regularization is

$$\frac{\partial^2 L}{\partial^2 U_i U_j} = \frac{\partial^2 L_0}{\partial^2 U_i U_j} + (1 - \delta_{ij}) 2\gamma_u (U_i + U_j) + \delta_{ij} 2\gamma_u; \tag{53}$$

$$\frac{\partial^2 L}{\partial^2 W_{jk}^i U_l} = \frac{\partial^2 L_0}{\partial^2 W_{jk}^i U_l} + 2(\gamma_w W_{jk}^i + \gamma_u U_l); \tag{54}$$

$$\frac{\partial^2 L}{\partial^2 W_{jk}^i W_{mn}^l} = \frac{\partial^2 L_0}{\partial^2 W_{jk}^i W_{mn}^l} + (1 - \delta_{il}\delta_{jm}\delta_{kn}) 2\gamma_w (W_{jk}^i + W_{mn}^l) + \delta_{il}\delta_{jm}\delta_{kn} 2\gamma_w. \tag{55}$$

For $U = 0$, $W = 0$, the Hessian of $L_0$ is 0, since each term in $L_0$ contains at least a $U$ or a $W$. The Hessian of $L$ becomes

$$\left. \frac{\partial^2 L}{\partial^2 U_i U_j} \right|_{U,W=0} = \delta_{ij} 2\gamma_u; \tag{56}$$

$$\left. \frac{\partial^2 L}{\partial^2 W_{jk}^i U_l} \right|_{U,W=0} = 0; \tag{57}$$

$$\left. \frac{\partial^2 L}{\partial^2 W_{jk}^i W_{mn}^l} \right|_{U,W=0} = \delta_{il}\delta_{jm}\delta_{kn} 2\gamma_w. \tag{58}$$

The Hessian of $L$ is a positive-definite matrix. Thus, $U = 0$, $W = 0$ is always a local minimum of the loss function $L$. $\square$

## B.8 Proof of Proposition 3

We first apply Lemma 3 to determine the condition for the nontrivial solution to exist. In particular, the Lemma must hold for $W^{(2)}$ and $W^{(1)}$, which leads to the following condition:

$$\|b^{D-1}d_0^{D-1}[b^{2D}d_0^D(\sigma^2+d_0)^D A_0 + \gamma]^{-1}\mathbb{E}[xy]\|^2 = 1. \tag{59}$$

Note that the left-hand side is a continuous function that tends to $0$ as $b \to \infty$. Therefore, it is sufficient to find the condition that guarantees that there exists $b$ such that the l.h.s. is larger than 1. For any $b$, the l.h.s. is a monotonically decreasing function of any eigenvalue of $A_0$, and so the following two inequalities hold:

$$\begin{cases} \|b^{D-1}d_0^{D-1}(b^{2D}d_0^D(\sigma^2+d_0)^D\sigma_x^2 + \gamma)^{-1}\mathbb{E}[xy]\| \leq \|b^{D-1}d_0^{D-1}(b^{2D}d_0^D(\sigma^2+d_0)^D a_{\min} + \gamma)^{-1}\mathbb{E}[xy]\| \\ \|b^{D-1}d_0^{D-1}(b^{2D}d_0^D(\sigma^2+d_0)^D\sigma_x^2 + \gamma)^{-1}\mathbb{E}[xy]\| \geq \|b^{D-1}d_0^{D-1}(b^{2D}d_0^D(\sigma^2+d_0)^D a_{\max} + \gamma)^{-1}\mathbb{E}[xy]\|. \end{cases} \tag{60}$$

The second inequality implies that if

$$\|b^{D-1}d_0^{D-1}[b^{2D}d_0^D(\sigma^2+d_0)^D a_{\max} + \gamma]^{-1}\mathbb{E}[xy]\| > 1, \tag{61}$$

a nontrivial solution must exist. This condition is equivalent to the existence of a $b$ such that

$$d_0^D(\sigma^2+d_0)^D a_{\max}b^{2D} - \|\mathbb{E}[xy]\|b^{D-1}d_0^{D-1} < -\gamma, \tag{62}$$

which is a polynomial inequality that does not admit an explicit condition for $b$ for a general $D$. Since the l.h.s is a continuous function that increases to infinity as $b \to \infty$, one sufficient condition for (62) to hold is that the minimizer of the l.h.s. is smaller than $\gamma$.

Note that the left-hand side of Eq. (62) diverges to $\infty$ as $b \to \pm\infty$ and tends to zero as $b \to 0$. Moreover, Eq. (62) is lower-bounded and must have a nontrivial minimizer for some $b > 0$ because the coefficient of the $b^{D-1}$ term is strictly negative. One can thus find its minimizer by taking derivative. In particular, the left-hand side is minimized when

$$b^{D+1} = \frac{(D-1)\|\mathbb{E}[xy]\|}{2Dd_0(\sigma^2+d_0)^D a_{\max}}, \tag{63}$$

and we can obtain the following sufficient condition for (62) to be satisfiable, which, in turn, implies that (59) is satisfiable:

$$\frac{D+1}{2D}\|\mathbb{E}[xy]\|d_0^{D-1}\left(\frac{(D-1)\|\mathbb{E}[xy]\|}{2Dd_0(\sigma^2+d_0)^D a_{\max}}\right)^{\frac{D-1}{D+1}} > \gamma, \tag{64}$$

which is identical to the proposition statement in (15).

Now, we come back to condition (60) to derive a sufficient condition for the trivial solution to be the only solution. The first inequality in Condition (60) implies that if

$$\|b^{D-1}d_0^{D-1}[b^{2D}d_0^D(\sigma^2+d_0)^D a_{\min} + \gamma]^{-1}\mathbb{E}[xy]\| \leq 1, \tag{65}$$

the only possible solution is the trivial one, and the condition for this to hold can be found using the same procedure as above to be

$$\frac{D+1}{2D}\|\mathbb{E}[xy]\|d_0^{D-1}\left(\frac{(D-1)\|\mathbb{E}[xy]\|}{2Dd_0(\sigma^2+d_0)^D a_{\min}}\right)^{\frac{D-1}{D+1}} \leq \gamma, \tag{66}$$

which is identical to (14).

We now prove the upper bound for the solution in ((16)). Because for any $b$, the first condition in 60 gives an upper bound, and so any $b$ that makes the upper bound less than 1 cannot be a solution. This means that any $b$ for which

$$\|b^{D-1}d_0^{D-1}[b^{2D}d_0^D(\sigma^2+d_0)^D a_{\min} + \gamma]^{-1}\mathbb{E}[xy]\| \leq 1 \tag{67}$$

cannot be a solution. This condition holds if and only if

$$d_0^D(\sigma^2+d_0)^D a_{\min}b^{2D} - \|\mathbb{E}[xy]\|b^{D-1}d_0^{D-1} > -\gamma. \tag{68}$$

Because $\gamma > 0$, one sufficient condition to ensure this is that there exists $b$ such that

$$d_0(\sigma^2 + d_0)^D a_{\min} b^{2D} - \|\mathbb{E}[xy]\| b^{D-1} > 0, \tag{69}$$

which is equivalent to

$$b > \left[ \frac{\|\mathbb{E}[xy]\|}{d_0(\sigma^2 + d_0)^D a_{\min}} \right]^{\frac{1}{D+1}}. \tag{70}$$

Namely, any solution $b^*$ satisfies

$$b^* \leq \left[ \frac{\|\mathbb{E}[xy]\|}{d_0(\sigma^2 + d_0)^D a_{\min}} \right]^{\frac{1}{D+1}}. \tag{71}$$

We can also find a lower bound for all possible solutions. When $D > 1$, another sufficient condition for Eq. (68) to hold is that there exists $b$ such that

$$\|\mathbb{E}[xy]\| d_0^{D-1} b^{D-1} < \gamma. \tag{72}$$

because the $b^{2D}$ term is always positive. This condition then implies that any solution must satisfy:

$$b^* \geq \frac{1}{d_0} \left[ \frac{\gamma}{\|\mathbb{E}[xy]\|} \right]^{\frac{1}{D-1}}. \tag{73}$$

For $D = 1$, we have by Theorem 1 that

$$b^* > 0 \tag{74}$$

if and only if $\mathbb{E}[xy] > \gamma$. This means that

$$b^* \geq \lim_{\eta \to 0^+} \lim_{D \to 1^+} \frac{1}{d_0} \left[ \frac{\gamma + \eta}{\|\mathbb{E}[xy]\|} \right]^{\frac{1}{D-1}} = \begin{cases} \infty & \text{if } \mathbb{E}[xy] \geq \gamma; \\ 0 & \text{if } \mathbb{E}[xy] < \gamma. \end{cases} \tag{75}$$

This finishes the proof. □

### B.9 Proof of Theorem 3

*Proof.* When nontrivial solutions exist, we are interested in identifying when $b = 0$ is not the global minimum. To achieve this, we compare the loss of $b = 0$ with the other solutions. Plug the trivial solution into the loss function in Eq. (9), the loss is easily identified to be $L_{\text{trivial}} = E[y^2]$.

For the nontrivial minimum, defining $f$ to be the model,

$$f(x) := \sum_{i,i_1,i_2,\ldots,i_D}^{d,d_1,d_2,\ldots d_D} U_{i_D} \epsilon_{i_D}^{(D)} \ldots \epsilon_{i_2}^{(2)} W_{i_2 i_1}^{(2)} \epsilon_{i_1}^{(1)} W_{i_1 i}^{(1)} x \tag{76}$$

$$= \eta d_0^D b^{2D} \mathbb{E}[xy]^T [b^{2D} d_0^D (\sigma^2 + d_0)^D A_0 + \gamma I]^{-1} x, \tag{77}$$

where, similar to the previous proof, we have defined $\sum_{i_1,\ldots,i_D}^{d_1,\ldots d_D} \epsilon_{i_D}^{(D)} \ldots \epsilon_{i_1}^{(1)} := \eta$ such that $\mathbb{E}[\eta] = \prod_i^D d_i = d_0^D$ and $\mathbb{E}[\eta^2] = \prod_i^D d_i(\sigma_i^2 + d_i) := d_0^D(\sigma^2 + d_0)^D$. With this notation, The loss function becomes

$$\mathbb{E}_x \mathbb{E}_\eta (f(x) - y)^2 + L_2 \ reg. \tag{78}$$

$$= \mathbb{E}_{x,\eta}[f(x)^2] - 2\mathbb{E}_{x,\eta}[yf(x)] + \mathbb{E}_x[y^2] + L_2 \ reg. \tag{79}$$

$$= \sum_i \frac{d_0^{3D}(\sigma^2 + d_0)^D b^{4D} a_i \mathbb{E}[x'y]_i^2}{[d_0^D(\sigma^2 + d_0)^D a_i b^{2D} + \gamma]^2} - 2 \sum_i \frac{d_0^{2D} b^{2D} \mathbb{E}[x'y]_i^2}{d_0^D(\sigma^2 + d_0)^D a_i b^{2D} + \gamma} + \mathbb{E}_x[y^2] + L_2 \ reg. \tag{80}$$

The last equation is obtained by rotating $x$ using a orthogonal matrix such that $R^{-1} A_0 R = \text{diag}(a_i)$ and denoting the rotated $x$ as $x' = Rx$. With $x'$, The $L_2 \ reg$ term takes the form of

$$L_2 \ reg. = \gamma D d_0^2 b^2 + \gamma \sum_i \frac{d_0^{2D} b^{2D} \|\mathbb{E}[x'y]_i\|^2}{(d_0^D(\sigma^2 + d_0)^D b^{2D} a_i + \gamma)^2}. \tag{81}$$

Combining the expressions of (81) and (80), we obtain that the difference between the loss at the non-trivial solution and the loss at 0 is

$$-\sum_i \frac{d_0^{2D} b^{2D} \mathbb{E}[x'y]_i^2}{[d_0^D(\sigma^2+d_0)^D a_i b^{2D}+\gamma]} + \gamma D d_0^2 b^2. \tag{82}$$

Satisfaction of the following relation thus guarantees that the global minimum is nontrivial:

$$\sum_i \frac{d_0^{2D} b^{2D} \mathbb{E}[x'y]_i^2}{[d_0^D(\sigma^2+d_0)^D a_i b^{2D}+\gamma]} \geq \gamma D d_0^2 b^2. \tag{83}$$

This relation is satisfied if

$$\frac{d_0^{2D} b^{2D} \|\mathbb{E}[xy]\|^2}{[d_0^D(\sigma^2+d_0)^D a_{max} b^{2D}+\gamma]} \geq \gamma D d_0^2 b^2 \tag{84}$$

$$\frac{b^{2D-2}}{[d_0^D(\sigma^2+d_0)^D a_{max} b^{2D}+\gamma]} \geq \frac{\gamma D}{d_0^{2D-2}\|\mathbb{E}[xy]\|^2}. \tag{85}$$

$$\tag{86}$$

The derivative or l.h.s. with respect to $b$ is

$$\frac{b^{2D-3}[(2D-2)\gamma - 2d_0^D(\sigma^2+d_0)^D a_{max} d^{2D}]}{[d_0^D(\sigma^2+d_0)^D a_{max} b^{2D}+\gamma]^2}. \tag{87}$$

For $b,\gamma \in (0,\infty)$, the derivative dives below 0, indicating the l.h.s. of (86) has a global maximum at a strictly positive $b$. The value of $b$ is found when setting the derivative to 0, namely

$$\frac{b^{2D-3}[(2D-2)\gamma - 2d_0^D(\sigma^2+d_0)^D a_{max} d^{2D}]}{[d_0^D(\sigma^2+d_0)^D a_{max} b^{2D}+\gamma]^2} = 0 \tag{88}$$

$$(2D-2)\gamma - 2d_0^D(\sigma^2+d_0)^D a_{max} d^{2D} = 0 \tag{89}$$

$$b^{2D} = \frac{(D-1)\gamma}{d_0^D(\sigma^2+d_0)^D a_{max}}. \tag{90}$$

The maximum value then takes the form

$$\frac{(D-1)^{\frac{D-1}{D}}}{D\gamma^{\frac{1}{D}} d_0^{D-1}(\sigma^2+d_0)^{D-1} a_{max}^{\frac{D-1}{D}}}. \tag{91}$$

The following condition thus guarantees that the global minimum is non-trivial

$$\frac{(D-1)^{\frac{D-1}{D}}}{D\gamma^{\frac{1}{D}} d_0^{D-1}(\sigma^2+d_0)^{D-1} a_{max}^{\frac{D-1}{D}}} \geq \frac{\gamma D}{d_0^{2D-2}\|\mathbb{E}[xy]\|^2} \tag{92}$$

$$\|\mathbb{E}[xy]\|^2 \geq \frac{\gamma^{\frac{D+1}{D}} D^2(\sigma^2+d_0)^{D-1} a_{max}^{\frac{D-1}{D}}}{d_0^{D-1}(D-1)^{\frac{D-1}{D}}}. \tag{93}$$

This finishes the proof. □

## B.10  Proof of Theorem 4

*Proof.* The model prediction is:

$$f(x) := \sum_{i,i_1,i_2,\ldots,i_D}^{d,d_1,d_2,\ldots d_D} U_{i_D}\epsilon_{i_D}^{(D)}\ldots\epsilon_{i_2}^{(2)} W_{i_2 i_1}^{(2)}\epsilon_{i_1}^{(1)} W_{i_1 i}^{(1)} x \tag{94}$$

$$= \eta d_0^D b^{2D} \mathbb{E}[xy]^T [b^{2D} d_0^D(\sigma^2+d_0)^D \sigma_x^2 I + \gamma I]^{-1} x. \tag{95}$$

One can find the expectation value and variance of a model prediction:

$$\mathbb{E}_\eta[f(x)] = \frac{d_0^{2D} b^{2D} \mathbb{E}[xy]^T x}{b^{2D} d_0^D(\sigma^2+d_0)^D \sigma_x^2 + \gamma} \tag{96}$$

For the trivial solution, the theorem is trivially true. We thus focus on the case when the global minimum is nontrivial.

The variance of the model is

$$Var[f(x)] = \mathbb{E}[f(x)^2] - \mathbb{E}[f(x)]^2 \tag{97}$$

$$= \frac{(\sigma^2 + d_0)^D d_0^{3D} b^{4D} (\mathbb{E}[xy]^T x)^2}{[b^{2D} d_0^D (\sigma^2 + d_0)^D \sigma_x^2 + \gamma]^2} - \frac{d_0^{4D} b^{4D} (\mathbb{E}[xy]^T x)^2}{[b^{2D} d_0^D (\sigma^2 + d_0)^D]^2 \sigma_x^2 + \gamma]^2} \tag{98}$$

$$= \frac{d_0^{3D}[(\sigma^2 + d_0)^D - d_0^D] b^{4D} (\mathbb{E}[xy]^T x)^2}{[b^{2D} d_0^D (\sigma^2 + d_0)^D \sigma_x^2 + \gamma]^2} \tag{99}$$

$$= \frac{d_0^{3D}[(\sigma^2 + d_0)^D - d_0^D] b^{2D+2} (\mathbb{E}[xy]^T x)^2}{\|\mathbb{E}[xy]\|^2}, \tag{100}$$

where the last equation follows from Eq. (13). The variance can be upper-bounded by applying (16),

$$Var[f(x)] \leq \frac{d_0^D[(\sigma^2 + d_0)^D - d_0^D](\mathbb{E}[xy]^T x)^2}{(\sigma^2 + d_0)^{2D} \sigma_x^2} \propto \frac{d_0^D[(\sigma^2 + d_0)^D - d_0^D]}{(\sigma^2 + d_0)^{2D}}. \tag{101}$$

We first consider the limit $d_0 \to \infty$ with fixed $\sigma^2$:

$$Var[f(x)] \propto \frac{D d_0^{2D-1} \sigma^2}{(d_0 + \sigma^2)^{2D}} = O\left(\frac{1}{d_0}\right). \tag{102}$$

For the limit $\sigma^2 \to \infty$ with $d_0$ fixed, we have

$$Var[f(x)] = O\left(\frac{1}{(\sigma^2)^D}\right). \tag{103}$$

Additionally, we can consider the limit when $D \to \infty$ as we fix both $\sigma^2$ and $d_0$:

$$Var[f(x)] = O\left(e^{-D 2 \log[(\sigma^2 + d_0)/d_0]}\right), \tag{104}$$

which is an exponential decay. □

## C    Exact Form of $b^\star$ for $D = 1$

Note that our main result does not specify the exact value of $b^\star$. This is because $b^\star$ must satisfy the condition in Eq. (6), which is equivalent to a high-order polynomial in $b$ with coefficients being general functions of the eigenvalues of $A_0$, whose solutions are generally not analytical by Galois theory. One special case where an analytical formula exists for $b$ is when $A_0 = \sigma_x^2 I$. Practically, this can be achieved for any (full-rank) dataset if we disentangle and rescale the data by the whitening transformation: $x \to \sigma_x \sqrt{A_0^{-1}} x$. In this case, we have

$$b_\star^2 = \frac{\sqrt{\frac{\gamma_w}{\gamma_u}} \|\mathbb{E}[xy]\| - \gamma_w}{(\sigma^2 + d_1)\sigma_x^2}, \tag{105}$$

and

$$v = \pm \sqrt{\frac{\sqrt{\frac{\gamma_u}{\gamma_w}} \|\mathbb{E}[xy]\| - \gamma_u}{\sigma_x^2 (\sigma^2 + d_1)}} \frac{\mathbb{E}[xy]}{\|\mathbb{E}[xy]\|}, \tag{106}$$

where $v = W_{i:}$.

## D    Effect of Bias

This section studies a deep linear network with biases for every layer and compares it with the no-bias networks. We first study a general case when the data does not receive any preprocessing. We then show that the problem reduces to the setting we considered in the main text under the common data preprocessing schemes that centers the input and output data: $\mathbb{E}[x] = 0$, and $\mathbb{E}[y] = 0$.

### D.1 Two-layer network

The two-layer linear network with bias is defined as

$$f_b(x; U, W, \beta^U, \beta^W) = \sum_i \epsilon_i U_i(W_{i:} \cdot x + \beta_i^W) + \beta^U, \tag{107}$$

where $\beta^W \in \mathbb{R}^{d_1}$ is the bias in the hidden layer, and $\beta^U \in \mathbb{R}$ is the bias at the output layer. The loss function is

$$L_b(U, W, \beta^U, \beta^W) = \mathbb{E}_{\epsilon,x,y}\Big[\big(\sum_i \epsilon_i U_i(W_{i:} \cdot x + \beta_i^W) + \beta^U - y\big)^2\Big] + L_2 \tag{108}$$

$$= \mathbb{E}_{x,y}\Big[(UWx + U\beta^W + \beta^U - y)^2 + \sigma^2 \sum_i U_i^2(W_{i:} \cdot x + \beta_i^W)^2\Big] \tag{109}$$

$$+ \gamma_u(\|U\|^2 + (\beta^U)^2) + \gamma_w(\|W\|^2 + \|\beta^W\|^2). \tag{110}$$

It is helpful to concatenate $x$ and 1 into a single vector $x' := (x, 1)^{\mathrm{T}}$ and concatenate $W$ and $\beta^W$ into a single matrix $W'$ such that $W$, $\beta^W$, $x$, and $W'$, $x'$ are related via the following equation

$$Wx + \beta^W = W'x'. \tag{111}$$

Using $W'$ and $x'$, the model can be written as

$$f_b(x', U, W', \beta^U) = \sum_i \epsilon_i U_i W'_{i:} \cdot x' + \beta^U. \tag{112}$$

The loss function simplifies to

$$L_b(U, W', \beta) = \mathbb{E}_{\epsilon,x,y}\Big[\big(\sum_i \epsilon_i U_i W'_{i:} \cdot x' + \beta^U - y\big)^2\Big] + \gamma_u(\|U\|^2 + (\beta^U)^2) + \gamma_w\|W'\|^2. \tag{113}$$

Note that (113) contains similar rescaling invariance between $U$ and $W'$ and the invariance of aligning $W'_{i:}$ and $W'_{j:}$. One can thus obtain the following two propositions that mirror Lemma 1 and 2.

**Proposition 5.** *At the global minimum of* (108), $U_j^2 = \frac{\gamma_w}{\gamma_u}\big(\sum_i W_{ji}^2 + (\beta_j^W)^2\big)$.

**Proposition 6.** *At the global minimum, for all $i$ and $j$, we have*

$$\begin{cases} U_i^2 = U_j^2; \\ U_i W_{i:} = U_j W_{j:}; \\ U_i \beta_i^W = U_j \beta_j^W. \end{cases} \tag{114}$$

The proofs are omitted because they are the same as those of Lemma 1 and 2, substituting $W$ by $W'$. By following a procedure similar to finding the solution for a no-bias network, one finds that

**Theorem 5.** *The global minimum of Eq.* (108) *is of the form*

$$\begin{cases} U = b\mathbf{r}; \\ \beta^U = \frac{d_1 b\big[\big((d_1+\sigma^2)\frac{\gamma_u}{\gamma_w}b-1\big)v\mathbb{E}[x] - \big(d_1\frac{\gamma_u}{\gamma_w}b^2-1\big)\mathbb{E}[y]\big]}{(d_1+\sigma^2)d_1\frac{\gamma_u^2}{\gamma_w^2}b^4 + (\gamma_u-2)d_1\frac{\gamma_u}{\gamma_w}b^2 + \gamma_u+1}; \\ W = \mathbf{r}b\Big\{\mathbb{E}[x]\Big[b\frac{\gamma_u}{\gamma_w}(d_1+b\sigma^2)-1\Big]\beta^U + \mathbb{E}[xy]\Big\}^T\big[b^2(d_1+\sigma^2)A_0 + \gamma_w I\big]^{-1}; \\ \beta^W = -\mathbf{r}\frac{\gamma_u}{\gamma_w}b\beta^U, \end{cases} \tag{115}$$

*where $b$ satisfies*

$$\gamma_u b^2 = b^2 \frac{\gamma_w(\frac{\mathbb{E}[y]S_1 S_3}{S_4}\mathbb{E}[x] - \mathbb{E}[xy])(M^{-1})^2(\frac{\mathbb{E}[y]S_1 S_3}{S_4}\mathbb{E}[x] - \mathbb{E}[xy])^T + \frac{\gamma_u^2}{\gamma_w}\big(\frac{S_3}{S_4}\mathbb{E}[y] - b\frac{S_2}{S_4}\mathbb{E}[x]M^{-1}\mathbb{E}[xy]\big)^2}{\big(b\frac{S_2 S_1}{S_4}\mathbb{E}[x]M^{-1}\mathbb{E}[x]^T - 1\big)^2}, \tag{116}$$

*where $M, S_1, S_2, S_3, S_4$ are functions of the model parameters and $b$, defined in Eq.* (122).

*Proof.* First of all, we derive a handy relation satisfied by $\beta^U$ and $\beta^W$ at all the stationary points. The zero-gradient condition of the stationary points gives

$$
\begin{cases}
\mathbb{E}_{x,y}[2(UWx + U\beta^W + \beta^U - y)]U + 2\gamma_w \beta^W = 0; \\
\mathbb{E}_{x,y}[2(UWx + U\beta^W + \beta^U - y)] + 2\gamma_u \beta^U = 0,
\end{cases}
\tag{117}
$$

leading to

$$
U\gamma_u \beta^U + \gamma_w \beta^W = 0
\tag{118}
$$

$$
\beta_i^W = -\frac{\gamma_u}{\gamma_w} U_i \beta^U.
\tag{119}
$$

Proposition 5 and proposition 6 implies that we can define $b := |U_i|$ and $bv := U_i W_{i:}$. Consequently, $U_i \beta_i^W = -\frac{\gamma_u}{\gamma_w} b^2 \beta^U$, and the loss function can be written as

$$
\mathbb{E}_x \left[ \left( bd_1 \sum_j v_j x_j - (d_1 \frac{\gamma_u}{\gamma_w} b^2 - 1)\beta^U - y \right)^2 + b^2 d_1 \sigma^2 \left( \sum_k v_k x_k - \frac{\gamma_u}{\gamma_w} b\beta^U \right)^2 \right] + \gamma_u d_1 b^2
$$

$$
+ \gamma_w d_1 \|v\|_2^2 + \gamma_u \left( \frac{b^2 d_1 \gamma_u}{\gamma_w} + 1 \right)(\beta^U)^2.
\tag{120}
$$

The respective zero-gradient condition for $v$ and $\beta^U$ implies that for all stationary points,

$$
\begin{cases}
v = [b^2(d_1 + \sigma^2)A_0 + \gamma_w I]^{-1} b \left\{ \mathbb{E}[x] \left[ b\frac{\gamma_u}{\gamma_w}(d_1 + b\sigma^2) - 1 \right]\beta^U + \mathbb{E}[xy] \right\}; \\
\beta^U = \frac{d_1 b\left[(d_1+\sigma^2)\frac{\gamma_u}{\gamma_w}b - 1\right]v\mathbb{E}[x] - \left(d_1 \frac{\gamma_u}{\gamma_w}b^2 - 1\right)\mathbb{E}[y]}{(d_1+\sigma^2)d_1 \frac{\gamma_u^2}{\gamma_w^2}b^4 + (\gamma_u - 2)d_1 \frac{\gamma_u}{\gamma_w}b^2 + \gamma_u + 1}.
\end{cases}
\tag{121}
$$

To shorten the expressions, we introduce

$$
\begin{cases}
M = b^2(d_1 + \sigma^2)A_0 + \gamma_w I; \\
S_1 = b\frac{\gamma_u}{\gamma_w}(d_1 + b\sigma^2) - 1; \\
S_2 = d_1 b\left[(d_1 + \sigma^2)\frac{\gamma_u}{\gamma_w}b - 1\right]; \\
S_3 = d_1 \frac{\gamma_u}{\gamma_w}b^2 - 1; \\
S_4 = (d_1 + \sigma^2)d_1 \frac{\gamma_u^2}{\gamma_w^2}b^4 + (\gamma_u - 2)d_1 \frac{\gamma_u}{\gamma_w}b^2 + \gamma_u + 1.
\end{cases}
\tag{122}
$$

With $M, S_1, S_2, S_3, S_4$, we have

$$
\begin{cases}
v = M^{-1}b(\mathbb{E}[x]S_1 \beta^U + \mathbb{E}[xy]); \\
\beta^U = \frac{S_2 v\mathbb{E}[x] - S_3 \mathbb{E}[y]}{S_4}.
\end{cases}
\tag{123}
$$

The inner product of $v$ and $\mathbb{E}[x]$ can be solved as

$$
v\mathbb{E}[x] = b\frac{\frac{S_3}{S_4}\mathbb{E}[x]M^{-1}\mathbb{E}[x]S_1\mathbb{E}[y] - \mathbb{E}[x]M^{-1}\mathbb{E}[xy]}{b\frac{S_2}{S_4}\mathbb{E}[x]M^{-1}\mathbb{E}[x]S_1 - 1}.
\tag{124}
$$

Inserting the expression of $v\mathbb{E}[x]$ into the expression of $\beta^U$ one obtains

$$
\beta^U = \frac{S_3 \mathbb{E}[y] - bS_2 \mathbb{E}[x]M^{-1}\mathbb{E}[xy]}{b\mathbb{E}[x]M^{-1}\mathbb{E}[x]S_1 S_2 - S_4}
\tag{125}
$$

The global minimum must thus satisfy

$$
\gamma_u b^2 = \gamma_w \|v\|^2 + \gamma_u \frac{b^2 d_1 \gamma_u}{\gamma_w}(\beta^U)^2
\tag{126}
$$

$$
= b^2 \frac{\gamma_w(\frac{\mathbb{E}[y]S_1 S_3}{S_4}\mathbb{E}[x] - \mathbb{E}[xy])(M^{-1})^2(\frac{\mathbb{E}[y]S_1 S_3}{S_4}\mathbb{E}[x] - \mathbb{E}[xy])^T + \frac{\gamma_u^2}{\gamma_w}\left(\frac{S_3}{S_4}\mathbb{E}[y] - b\frac{S_2}{S_4}\mathbb{E}[x]M^{-1}\mathbb{E}[xy]\right)^2}{\left(b\frac{S_2 S_1}{S_4}\mathbb{E}[x]M^{-1}\mathbb{E}[x]^T - 1\right)^2}.
\tag{127}
$$

This completes the proof. □

**Remark.** *As in the no-bias case, we have reduced the original problem to a one-dimensional problem. However, the condition for b becomes so complicated that it is almost impossible to understand. That being said, the numerical simulations we have done all carry the bias terms, suggesting that even with the bias term, the mechanisms are qualitatively similar, and so the approach in the main text is justified.*

When $\mathbb{E}[x] = 0$, the solution can be simplified a little:

$$
\begin{cases}
U = \mathbf{r}b; \\
\beta^U = -\dfrac{d_1 \frac{\gamma_u}{\gamma_w} b^2 - 1}{(d_1+\sigma^2)d_1 \frac{\gamma_u^2}{\gamma_w^2} b^4 + (\gamma_u - 2)d_1 \frac{\gamma_u}{\gamma_w} b^2 + \gamma_u + 1} \mathbb{E}[y]; \\
W = \mathbf{r}b\mathbb{E}[xy]^T [b^2(d_1 + \sigma^2)A_0 + \gamma_w I]^{-1}; \\
\beta^W = \mathbf{r}\dfrac{\gamma_u}{\gamma_w}b \dfrac{d_1 \frac{\gamma_u}{\gamma_w} b^2 - 1}{(d_1+\sigma^2)d_1 \frac{\gamma_u^2}{\gamma_w^2} b^4 + (\gamma_u - 2)d_1 \frac{\gamma_u}{\gamma_w} b^2 + \gamma_u + 1} \mathbb{E}[y],
\end{cases}
\tag{128}
$$

where the value of $b$ is either $0$ or determined by

$$
\gamma_u = \gamma_w |\mathbb{E}[xy]^T [b^2(d_1+\sigma^2)A_0 + \gamma_w I]^{-1}|^2 + \frac{\gamma_u^2}{\gamma_w}\mathbb{E}[y]^2 \left( \frac{d_1 \frac{\gamma_u}{\gamma_w} b^2 - 1}{(d_1 + \sigma^2)d_1 \frac{\gamma_u^2}{\gamma_w^2} b^4 + (\gamma_u - 2)d_1 \frac{\gamma_u}{\gamma_w} b^2 + \gamma_u + 1} \right)^2 .
\tag{129}
$$

In this case, the expression of $W$ is identical to the no-bias model. The bias of both layers is proportional to $\mathbb{E}[y]$. The equation determining the value of $b$ is also similar to the no-bias case. The only difference is the term proportional to $\mathbb{E}[y]^2$.

We also note that the solution becomes significantly simplified when $\mathbb{E}[x] = 0$ and $\mathbb{E}[y] = 0$. This could be seen by finding the partial derivative of $L$ with respect to $\beta^W$ and $\beta^U$ and then setting them to $0$. When $\mathbb{E}[x] = 0$, $\mathbb{E}[y] = 0$, one obtains:

$$
\begin{cases}
\frac{\partial L}{\partial \beta_i^W} = U_i U \beta^W + U_i \beta^U + \gamma_w \beta_i^W = 0; \\
\frac{\partial L}{\partial \beta^U} = U \beta^W + \beta^U + \gamma_u \beta^U = 0.
\end{cases}
\tag{130}
$$

These equations lead to

$$
U \gamma_u \beta^U + \gamma_w \beta^W = 0,
\tag{131}
$$

implying

$$
\begin{cases}
\beta^U = 0; \\
\beta^W = 0.
\end{cases}
\tag{132}
$$

In practice, it is common and usually recommended practice to subtract the average of $x$ and $y$ from the data and achieve precisely $\mathbb{E}[x] = 0$ and $\mathbb{E}[y] = 0$. We generalize this result to deeper networks in the next section.

## D.2 Deep linear network

Let $\beta$ be a $\left(\sum_i^D d_i + 1\right)$-dimensional vector concatenating all $\beta^1, \beta^2, ..., \beta^D, \beta^U$, and denoting the collection of all the weights $U, W^D, ..., W^1$ by $w$, the model of a deep linear network with bias is defined as

$$
f_b(x, W^D, ..., W^1, U, \beta^D, ..., \beta^1, \beta^U)
\tag{133}
$$
$$
=(\epsilon^U \circ U)((\epsilon^D \circ W^D)(...((\epsilon^2 \circ W^2)((W^1 x + \beta^1) + \beta^2)...) + \beta^D) + \beta^U
\tag{134}
$$
$$
=(\epsilon^U \circ U)(\epsilon^D \circ W^D)...(\epsilon^2 \circ W^2)W^1 x + (\epsilon^U \circ U)(\epsilon^D \circ W^D)...(\epsilon^2 \circ W^2)\beta^1
\tag{135}
$$
$$
+ (\epsilon^U \circ U)(\epsilon^D \circ W^D)...(\epsilon^3 \circ W^3)\beta^2 + ... + (\epsilon^U \circ U)\beta^D + \beta^U
\tag{136}
$$
$$
=(\epsilon^U \circ U)(\epsilon^D \circ W^D)...(\epsilon^2 \circ W^2)W^1 x + bias(w, \beta),
\tag{137}
$$

where

$$
bias(w, \beta) = (\epsilon^U \circ U)(\epsilon^D \circ W^D)...(\epsilon^2 \circ W^2)\beta^1 + (\epsilon^U \circ U)(\epsilon^D \circ W^D)...(\epsilon^3 \circ W^3)\beta^2 + ... + (\epsilon^U \circ U)\beta^D + \beta^U,
\tag{138}
$$

and ∘ denotes Hadamard product. The loss function is

$$L_b(x, y, w, \beta) = \mathbb{E}_{\epsilon,x,y}[(f_b(x, w, \beta) - y)^2] + L_2(w, \beta). \tag{139}$$

Proposition 5 and Proposition 6 can be generated to deep linear network. Similar to the no-bias case, we can reduce the landscape to a 1-dimensional problem by performing induction on $D$ and using the 2-dimensional case as the base step. However, we do not solve this case explicitly here because the involved expressions now become too long and complicated even to write down, nor can they directly offer too much insight. We thus only focus on the case when the data has been properly preprocessed. Namely, $\mathbb{E}[x] = 0$ and $\mathbb{E}[y] = 0$.

For simplicity, we assume that the regularization strength for all the layers employs the value $\gamma$. The following theorem shows that When $\mathbb{E}[x] = 0$ and $\mathbb{E}[y] = 0$, the biases vanish for an arbitrarily deep linear network:

**Theorem 6.** *Let $\mathbb{E}[x] = 0$ and $\mathbb{E}[y] = 0$. The global minima of Eq. (139) have $\beta^1 = 0, \beta^2 = 0, ..., \beta^D = 0, \beta^U = 0$.*

*Proof.* At the global minimum, the gradient of the loss function vanishes. In particular, the derivatives with respect to $\beta$ vanish:

$$\frac{\partial L_b(x, y, w, \beta)}{\partial \beta_i} = 0; \tag{140}$$

$$\mathbb{E}_{\epsilon,x,y}\left[\frac{\partial f_b(x, w, \beta)}{\partial \beta_i}(f_b(x, w, \beta) - y)\right] + \gamma\beta_i = 0; \tag{141}$$

$$\mathbb{E}_{\epsilon,x,y}\left[\frac{\partial bias(w, \beta)}{\partial \beta_i}(f_b(x, w, \beta) - y)\right] + \gamma\beta_i = 0; \tag{142}$$

$$\mathbb{E}_{\epsilon}\left[\frac{\partial bias(w, \beta)}{\partial \beta_i}(f_b(\mathbb{E}[x], w, \beta) - \mathbb{E}[y])\right] + \gamma\beta_i = 0, \tag{143}$$

where $\beta_i$ is the $i$th element of $\beta$. The last equation is obtained since $f_b(x, w, \beta)$ is a linear function of $x$. Using the condition $\mathbb{E}[x] = 0$ and $\mathbb{E}[y] = 0$, Equation (143) becomes

$$\mathbb{E}_{\epsilon}\left[\frac{\partial bias(w, \beta)}{\partial \beta_i}bias(w, \beta)\right] + \gamma\beta_i = 0. \tag{144}$$

$bias(w, \beta)$ is a linear combination of $\beta_i$. Consequently, $\partial bias(w, \beta)/\partial \beta_i$ does not depend on $\beta$, and $bias(w, \beta)\partial bias(w, \beta)/\partial \beta_i$ is a linear combination of $\beta_i$. The $\sum_i^D d_i + 1$ equation derived from vanishing gradient yield a set of $\sum_i^D d_i + 1$ linear equations of the form $M(w)\beta = 0$, where $M(w)$ is a $\left(\sum_i^D d_i + 1\right) \times \left(\sum_i^D d_i + 1\right)$ matrix with dependence on $w$. These linear equations are linearly independent, since the term $\partial L_2(w, \beta)/\partial \beta_i = 2\gamma\beta_i$ and is different in each of the equations. Thus, the linear system $M(w)\beta = 0$ has $\left(\sum_i^D d_i + 1\right)$ independent equations and $\left(\sum_i^D d_i + 1\right)$ variables. The only possible solution to this linear system is

$$\beta = 0. \tag{145}$$

This finishes the proof. □

Thus, for a deep linear network, a model without bias is good enough to describe data satisfying $\mathbb{E}[x] = 0$ and $\mathbb{E}[y] = 0$, which could be achieved by subtracting the mean of the data.