# OpenReview forum: "Exact Solutions of a Deep Linear Network"
_NeurIPS.cc/2022/Conference — NeurIPS 2022 Accept_

### Official Review · Reviewer_QUdB · 2022-07-10

**Rating:** 7
**Confidence:** 4
**Soundness:** 4 excellent
**Presentation:** 3 good
**Contribution:** 3 good

**Summary:**

This paper studies deep linear neural networks with weight decay and stochastic neurons. The authors show that the analytical global minima of square loss can be found for shallow neural networks (thm1) and deep neural networks (thm2). The analysis has some implications on the role of weight decay and the depth of neural networks.

**Questions:**

The paper assumes $\mathbb{E}[\epsilon_i] = 1$ and $\mathbb{E}[\epsilon_i\epsilon_j] = \delta_{ij}\sigma^2 +1$ with $\sigma^2>0$. If $\epsilon$ are all 1, which is the case of neural networks with deterministic neurons, then $\sigma^2$ = 0 which does not fit into this regime. Is it correct?

**Limitations:**

Yes.

**Strengths And Weaknesses:**

Strengths:

I like the setting in this paper, which is clean and simple but can manifest interesting properties of neural networks. The results are very interesting, especially the part where bad minima emerge with weight decay.

Weakness:
1) I understand the difficulty of analyzing global minima hence some assumptions are needed, e.g., diagonal A_0 for the exact form of b^* for the shallow neural networks., and single data. But some results are not easy to interpret, e.g., Thm2, Prop 3. Maybe the authors can provide more intuitions.

2) I have a minor concern that in the main contributions, point 4 seems irrelevant to this paper.

-- What's v in Eq.(8)?

---

> ### Author Response · Authors · 2022-08-02
> **Author Reply**
>
> Thank you for the review and for the summary. To improve our manuscript based on your feedback, we have updated the discussion to improve the intuitive understanding of our results. We also extended our analysis to the case when $\sigma^2=0$.
>
>
> > “I understand the difficulty of analyzing global minima hence some assumptions are needed, e.g., diagonal A_0 for the exact form of b^* for the shallow neural networks., and single data. But some results are not easy to interpret, e.g., Thm2, Prop 3. Maybe the authors can provide more intuitions.”:
> - We have updated the related discussions
>
> > “I have a minor concern that in the main contributions, point 4 seems irrelevant to this paper.”:
> - Point 4 discusses the implication of theorem 4, which could be interesting for those who study neural networks with stochasticity, such as a network with Monte-Carlo dropout, which has been extensively used for uncertainty estimation in deep learning [proceedings.mlr.press/v48/gal16.html](https://proceedings.mlr.press/v48/gal16.html).
>
>
>
> > “What's v in Eq.(8)?”
> - Thanks for pointing out this typo. It refers to the rows of the matrix $W$. Also, this part has been removed during revision to save space for including more empirical results.
>
> > “The paper assumes $\mathbb{E}[\epsilon_i] = 1$ and with $\mathbb{E}[\epsilon_i\epsilon_j] = \delta_{ij}\sigma^2 + 1$ with $\sigma^2 >0$. If $\epsilon$ are all 1, which is the case of neural networks with deterministic neurons, then $\sigma = 0$ which does not fit into this regime. Is it correct?”:
> - Thanks for raising this question. We have now extended our analysis to the case when $\sigma^2=0$ in the appendix section E.
> In short, when $\sigma>0$, the global minima we found have an averaging characteristic. Let us thus call this solution the “averaging solution.” When there is no noise, the averaging solutions we identified are still global minima, just that they no longer enumerate all the global minima. Let $W^{i+1}$ and $W^{i}$ be the averaging solution of the consecutive layers, and let $R$ be any rotation matrix, the set of all rotated solutions (and across all consecutive layers) $W^{i+1}R, R^{-1}W^{i}$ now constitutes a manifold of global minima. This is because when there is no noise, the original loss becomes invariant to such rotations (whereas the zero-noise case removes this invariance).

---

### Official Review · Reviewer_SZFK · 2022-07-10

**Rating:** 5
**Confidence:** 4
**Soundness:** 3 good
**Presentation:** 3 good
**Contribution:** 3 good

**Summary:**

This paper provides a closed-form solution (up to some constant) of the global minima of linear neural networks when trained using square loss and strictly positive weight decay. This result can be extended to the case when the neurons are stochastic and independent. The formulas for the global minima are directly or potentially related to the weight decay, depth, stochasticity of neurons, and signal strength from training data. The authors also used the characterizations of global minima to explain multiple phenomena happening in real neural network training, e.g., deeper networks are harder to optimize. This paper also provided variance analysis in the asymptotic limits of network hyperparameters and did small-scale experiments on synthetic data to validate their theoretical results in non-linear networks.

**Questions:**

- Could the authors provide more theoretical or empirical evidence connecting the loss landscape of linear networks and those of non-linear ones?

- Is it true that the weights of linear neural networks will almost always converge to the global/local minima characterized by the authors in this paper? Is it possible that the weights of the neural networks could converge to some points that are not captured in this paper, or even diverge?

- Can the proof techniques used in this paper be generalized to other settings? For example, when there are other kinds of regularizations or no regularizations at all.

- (Minor questions) Could the authors address the two questions I mentioned in the "Minor Comments" section above? In other words, could the authors provide the detailed settings of their experiments and the definition of the notation "$v$" in equation (8)?

**Limitations:**

The authors stated the assumptions for their theoretical results in the paper and had many discussions about the implications. It might be better if the authors could discuss more explicitly the limitations of this paper in the implication and conclusion sections.

This paper is mostly theoretical and focuses on a fundamental problem in general neural network training, and I do not see any immediate negative societal impact of this work.

**Strengths And Weaknesses:**

Strengths:

- This paper theoretically gives an analytical formula (up to some constant) for the global minima of linear neural networks trained with weight decay, and this formula works for deep linear networks and could be generalized to independent stochastic neurons. These analytical expressions provide opportunities to study the properties of these global minima of deep linear networks in detail.

- It is an interesting idea to connect the formula of the global minima of neural networks to various common phenomena in this field, e.g., the collapses in deep learning.

- This paper is generally well-written and well-structured. The notations used in this paper are mostly well-defined, and the intuitions and implications of the theoretical results are provided in the main text. These make this paper easy to understand.

- The theoretical proofs in this paper appear to be correct, and the related works are adequately cited.

Weaknesses:

- The theoretical results in this paper are all about linear networks, and the relationship between linear networks are non-linear ones seems somewhat unclear, so most of the conclusions in this paper might not translate directly to non-linear settings. This is my major concern about this paper. The authors claimed on Line 16 that the landscapes of linear networks are believed to well approximate that of non-linear ones, but this claim might be vague and need further explanation. The authors also did experiments with a small two-layer non-linear network on synthetic data, but the scale of this experiment is small, so it is unclear whether this result still holds for more general settings. It would be better if the authors could provide more theoretical or empirical evidence connecting the loss landscape of linear neural networks and those of non-linear ones.

- The characterizations of global minima might not be enough to characterize the training of neural networks, which depends on the properties of the entire loss landscape. For instance, it is possible that the weights of neural networks could diverge and it never reaches a minimum, and it is also possible that the weights converge to a bad local minimum or saddle point. It might be better if the authors could (theoretically or empirically) eliminate these possibilities and show that the network weights will always converge to the points that they characterized, i.e., either the global minima or the bad local minimum at 0.

- The proof techniques used in this paper seem to heavily rely on the existence of weight decay at all layers, making it hard to be generalized to other settings. Without weight decay, relationships like equation (13) will break and the characterizations of the local and global minima could become much more complicated.

- Some arguments made in this paper might be somewhat vague. For example, in Line 268, it might be unclear what the authors mean by "cannot learn the data".

Minor Comments:

- The details of the experiments the authors did to produce Figure 2 are missing. These details (e.g., how the data are generated, and what the hyperparameters are) could be important for interpreting the experimental results.

- The notation "$v$" on the left-hand side of equation (8) seems undefined. Should it be defined as some term in equation (5)?

Typos:

- Line 243, "are global and cannot generalize" -> "are global cannot generalize"
- Line 269, "two-layer net, and the existence" -> "two-layer net, the existence"

-------------------------------------------------
Update after author response:

I have read all other reviews and the authors' responses, and I decided to increase my score by 2. There are two main reasons why I increase my score:
1.  The authors have added empirical evidence (e.g., ResNet on CIFAR) to further relate the loss landscape of linear neural networks to non-linear ones.
2. The proof framework in this paper can be extended to more general settings with similar results, and the authors have provided theoretical results in more general settings, especially when there is no weight decay.

---

> ### Author Response · Authors · 2022-08-02
> **Author Reply Part 1**
>
> Thank you for your detailed read through and the review. We added new empirical and theoretical results in response to your review. However, we feel that some of your original criticism may be too harsh. We will explain this below. Please also let us know if there are any remaining problems, we will make updates accordingly in future versions.
>
> > The theoretical results in this paper are all about linear networks, and the relationship between linear networks are non-linear ones seems somewhat unclear, so most of the conclusions in this paper might not translate directly to non-linear settings. This is my major concern about this paper. The authors claimed on Line 16 that the landscapes of linear networks are believed to well approximate that of non-linear ones, but this claim might be vague and need further explanation. The authors also did experiments with a small two-layer non-linear network on synthetic data, but the scale of this experiment is small, so it is unclear whether this result still holds for more general settings. It would be better if the authors could provide more theoretical or empirical evidence connecting the loss landscape of linear neural networks and those of non-linear ones.
>
> - Thanks for this criticism. We have added an experiment on ResNet18 on CIFAR10 further demonstrate the relevance of our results to modern nonlinear models (see Section 5)
> - We also added empirical results that show that the landscape of linear nets can be either qualitatively or quantitatively similar to those of nonlinear nets (ReLU, tanh, Swish) close the origin (Section G)
> - Below, we will discuss in more detail how our theoretical result is relevant for more general settings that involve a nonlinear model. Here, we would like to point out that even restricting our result to linear models, it should not warrant a rejection because, on its own, theoretical studies of deep linear nets are regarded as an important and fundamental problem in the field. For example, see Kawaguchi (2016), Arora, and the well-known open question stated in [arxiv.org/abs/1412.0233](https://arxiv.org/abs/1412.0233). The importance of this problem per se is also agreed on by reviewers PsE5 and QUdB.
>
>
>
>
> > The characterizations of global minima might not be enough to characterize the training of neural networks, which depends on the properties of the entire loss landscape. For instance, it is possible that the weights of neural networks could diverge and it never reaches a minimum, and it is also possible that the weights converge to a bad local minimum or saddle point. It might be better if the authors could (theoretically or empirically) eliminate these possibilities and show that the network weights will always converge to the points that they characterized, i.e., either the global minima or the bad local minimum at 0.
>
> - We feel that you have misunderstood the main goal of our paper. While the optimization of neural networks is an important question, the analysis of the loss landscape of neural networks is also of fundamental importance. This philosophy is supported by other reviewers and by all the related works we cited. After all, the possibility that neural networks may not converge to global minima is not a sufficient reason to reject works that study the global minima of neural networks.
>
> > The proof techniques used in this paper seem to heavily rely on the existence of weight decay at all layers, making it hard to be generalized to other settings. Without weight decay, relationships like equation (13) will break and the characterizations of the local and global minima could become much more complicated.
>
> - Thanks for this question. We have now added Section D in the appendix to extend our theory to the case when $\gamma=0$ and showed how our results could be used for understanding the no-weight-decay case. In fact, Mianjy and Arora (2019) have studied the case without weight decay.
>
>
>
> > Some arguments made in this paper might be somewhat vague. For example, in Line 268, it might be unclear what the authors mean by "cannot learn the data".
> - Because zero is the global minimum, the model has not captured any data-specific feature, and so it has not “learned the data.”
>
>
> > The details of the experiments the authors did to produce Figure 2 are missing. These details (e.g., how the data are generated, and what the hyperparameters are) could be important for interpreting the experimental results.
>
> - We now include the experimental details in Section F.
>
> > The notation "$v$" on the left-hand side of equation (8) seems undefined. Should it be defined as some term in equation (5)?
>
> - Thanks for pointing out this typo. It refers to the rows of the matrix $W$. Also, this part has been removed during revision to save space for including more empirical results.

---

> ### Author Response · Authors · 2022-08-02
> **Author Reply Part 2**
>
> > Could the authors provide more theoretical or empirical evidence connecting the loss landscape of linear networks and those of non-linear ones?
> - Thanks for the question. We have now added a comparison of ResNet18 on CIFAR with a deep linear net with roughly similar depth. See Section 5. We also include empirical results that show that the landscape of linear nets can be either qualitatively or quantitatively similar to those of nonlinear nets (ReLU, tanh, Swish) close to the origin in Section G.
>
>
> > Is it true that the weights of linear neural networks will almost always converge to the global/local minima characterized by the authors in this paper? Is it possible that the weights of the neural networks could converge to some points that are not captured in this paper, or even diverge?
>
> -  In general, the weights can converge to other minima. However, our work is based on the philosophy that the global minima is the most important property of the loss landscape. The properties of the global minimum should precisely reflect the intention of the user/practitioner who writes this loss function down: when we use a loss function, it should be the case that we want the model to find the global minimum of this loss function. If some solution other than the global minimum is desired, we should add a constraint/regularization term to favor these properties. This is why the loss function is called the “objective.”
> - Additionally, a big problem in learning practice is that the loss landscape of deep learning is so complex that when writing down a loss function, the practitioner often has no understanding of the actual global minimum of it. Due to this ignorance, he or she might have written down a loss function whose global minimum does not really achieve his goal, and our work precisely points out how this could happen: for a deep linear net, using an improper level of weight decay can make zero the global minimum, which is almost never a desirable thing. Thus, our focus on the property of the global minimum constitutes a novel and significant contribution even for practices.
>
>
>
> > Can the proof techniques used in this paper be generalized to other settings? For example, when there are other kinds of regularizations or no regularizations at all.
> - Yes. They are still relevant for these cases. We added theoretical results on the case of zero weight decay ($\gamma=0$) in Section D and on the case of zero noise ($\sigma^2=0$) in Section E
>
> > (Minor questions) Could the authors address the two questions I mentioned in the "Minor Comments" section above? In other words, could the authors provide the detailed settings of their experiments and the definition of the notation "$v$" in equation (8)?
>
> - Yes, see our reply above.
>
> > The authors stated the assumptions for their theoretical results in the paper and had many discussions about the implications. It might be better if the authors could discuss more explicitly the limitations of this paper in the implication and conclusion sections.
>
> - We toned down our claims in the paper and added discussions explicitly cautioning the limitation of our results.

---

> > ### Comment · Reviewer_SZFK · 2022-08-09
> > **Increase the score by 2**
> >
> > Thank the authors for their detailed response and for providing much additional material in the updated draft. I have read all other reviews and the authors' responses, and I decided to increase my score by 2. I have updated the review to reflect the changes. As mentioned in the updated review, there are two main reasons why I increase my score:
> > 1.  The authors have added empirical evidence (e.g., ResNet on CIFAR) to further relate the loss landscape of linear neural networks to non-linear ones.
> > 2. The proof framework in this paper can be extended to more general settings with similar results, and the authors have provided theoretical results in more general settings, especially when there is no weight decay.

---

### Official Review · Reviewer_PsE5 · 2022-07-22

**Rating:** 7
**Confidence:** 3
**Soundness:** 3 good
**Presentation:** 3 good
**Contribution:** 3 good

**Summary:**

This work studies the population loss landscape of stochastic (e.g., in the sense of dropout) deep, linear, neural networks under 2-norm weight regularization. The key contribution of this work is the derivation of analytical expressions for the global minima of the aforementioned loss landscape, at least up to a scalar quantity. The implications of this result for training both linear and non-linear neural networks are discussed. In particular, this result illustrates how weight decay and depth can lead to a more challenging optimization problems, as well as the importance of the role of network initialization in avoiding basins of attraction around bad minima.

**Questions:**

- Lemma 2 if I have understood correctly, it essentially seems to say the output layer weight vector $U$ is of the form $c\textbf{v}$, where $\textbf{v} \in\{ \pm 1\}^d$ vector, likewise the the rows of $W$ are just signs of the same constant row vector. How does this compare to the case with no noise?

- The networks you discuss have no bias parameters, how straightforward would it be for you to extend these results and techniques to also cover this case?

- The remark after proposition 1 is useful. Can you provide examples of cases where a global minimum doesn't exist when you set a single regularizer weight to 0?

**Limitations:**

I think the authors are reasonably upfront about the limitations of their work, although I think they could perhaps add some suggested avenues for future works. I can't envisage how this work might have a negative societal impact.

**Strengths And Weaknesses:**

Originality: this paper continues the line of work analyzing linear neural networks. I am not a specialist concerning the study of linear networks, but the results appear, at least to the best of my knowledge, novel and interesting.

Quality and clarity: on the whole I think the paper is well organized, well written and clear. A few very minor suggestions in regard to the paper's presentation are as follows.

- I think upfront you could state your network architecture/ forward pass function more clearly / in terms of matrix vector product and give the dimensions of each of your parameter matrices.

- Notation wise both scalars and vectors use lower case characters which can be a bit confusing, perhaps using bold lower case characters for vectors might help.

- It might be helpful for the reader to restate statements of lemmas and theorems in the supplementary so they don't have to flick between.

- Line 475 in the supplementary "...we see that the left hand side \textit{is} larger..."


Significance: I think the extent to which understanding linear networks is important for understand nonlinear networks is not entirely clear. I still think it is important that we understand deep linear networks regardless however, and this work seems like a useful contribution.

---

> ### Author Response · Authors · 2022-08-02
> **Author Reply**
>
> Thank you for the review. In response to your questions, we significantly extended our theoretical results to consider the case of having a bias and the effect of having no noise.
>
> > Lemma 2 if I have understood correctly, it essentially seems to say the output layer weight vector $U$ is of the form $c\textbf{v}$, where $\textbf{v}\in \pm1^d$ vector, likewise the the rows of $W$ are just signs of the same constant row vector. How does this compare to the case with no noise?
> - Thanks for asking this question. We added section E in the Appendix to connect our result to the case when $\sigma^2=0$. In short, the noise (dropout) encourages an averaging effect, and this is why the rows of $W$ are copies of one another. Let us call this the “averaging solution.” When there is no noise at all, this averaging solution is still a global minimum, just that it no longer enumerates all the global minima. Namely, let $W^{i+1}$ and $W^{i}$ be the averaging solution of the consecutive layers, and let $R$ be any rotation matrix, the set of all rotated solutions (and across all consecutive layers) $W^{i+1}R$, $R^{-1}W^{i}$ constitutes a manifold of global minima. This is because the original loss becomes invariant to such rotations when there is no noise.
>
> > The networks you discuss have no bias parameters, how straightforward would it be for you to extend these results and techniques to also cover this case?
> - We now added another technical analysis in Section C to explicitly study the case where explicit bias is included. We divided into two cases:
> 1. The case when $\mathbb{E}[x]=0$ and $\mathbb{E}[y]=0$. We showed that the global minima always have zero bias, and so our results apply unchanged. While this case may seem restrictive, it is often the recommended practice to center the input and output data, and so it is very practically-relevant
> 2. The general case. Here, we showed that the same techniques we used in the main text still apply, but the self-consistent equation becomes much more complicated even for a two-layer net. We also described how this result could be directly extended to multilayer nets.
>
> > The remark after proposition 1 is useful. Can you provide examples of cases where a global minimum doesn't exist when you set a single regularizer weight to 0?
> - Consider a 1-hidden-layer model $f(x)=VWx$, and we only apply L2 reg to, say, $W$. For any solution $V$ and $W$, we can perform this transformation: $V\to aV$ and $W\to W/a$. Note that this leaves the model prediction unchanged, and so the MSE part of the loss is unchanged. However, as $a\to \infty$, the L2 reg part of loss monotonically decreases and only reaches a minimum when $a=\infty$. Thus, the global minimum is divergent or “nonexistent.”

---

> > ### Comment · Reviewer_PsE5 · 2022-08-03
> > **Useful new material in the appendices**
> >
> > Great, thanks for answering my questions. The additional appendices I think definitely strengthen the paper and answer some natural questions. I encourage the authors to continue their work and investigate further the implications of this study for nonlinear neural networks.

---

### Official Review · Reviewer_UdKB · 2022-07-25

**Rating:** 5
**Confidence:** 4
**Soundness:** 3 good
**Presentation:** 4 excellent
**Contribution:** 3 good

**Summary:**

This paper analyzes the global minima of deep linear networks with weight decay. Under the assumption of linear architecture, $l_2$ regularization and population risk, the paper takes advantage of the symmetry and invariance in the network and derives the analytical expression of the minimum points. Depending on the regularization strength, for two-layer networks, zero is either the global minimum or a saddle point; for deeper networks, zero is always a local minimum and can be global (see Figure 1). The paper also tries to connect these theoretical results with some phenomena in deep nonlinear networks.

**Questions:**

1. An important result of this paper is that zero is a saddle point (not local minimum) when $D = 1$ if the strength of the signal is more significant. It would be better to mention it in Theorem 1 or its discussion.
2. In Figure 1 Left, the labels of the inequality may need update.
3. In Figure 2 Right, does the linear regression also have l2 regularization?

**Limitations:**

Please see the “Weaknesses” Section above.

**Strengths And Weaknesses:**

Strengths:
1. The paper proposes a set of assumptions under which the analytical expression of the minimum points can be derived, and the corresponding properties can be analyzed.
2. I think the most interesting contribution of this paper is to point out that the weight decay, i.e., the $L_2$ regularization, may introduce a local minimum at zero for deep networks. It is not surprising since the regularization term in Eqn (1) is quadratic while $L_0$ has higher order. It is good to formulate the phenomenon as a rigorous theory under the assumptions.
3. The presentation of the paper is pretty good. The settings are clearly stated and the results are supported by rigorous proofs. There are also detailed comments and discussions about the meaning and possible implications of the theoretical results.

Weaknesses:
My main concern is that the assumptions in this paper may over simplified the problem. The proofs are straightforward (though I believe Theorem 2 is not trivial), and heavily depend on the symmetry (thanks to the assumptions) that does not hold in general cases. If considering weaker assumptions, I guess we may still prove that zero is a local minimum, since the regularization is quadratic and the square loss has higher order, but the quantitative results in this paper may not be extended. Since the implications in Section 5 are all under weaker assumptions, the theoretical results may not support the discussions here strongly enough. In addition, I guess a ResNet architecture may avoid the local minimum at zero since the square loss is not in high order now.

---

> ### Author Response · Authors · 2022-08-02
> **Author Reply**
>
> Thank you very much for the review. In response, we clarify the relevance of our result to more general settings and soften the tone of discussions in our paper to avoid misleading the reader.
>
> > My main concern is that the assumptions in this paper may over simplified the problem. The proof is straightforward (though I believe Theorem 2 is not trivial), and heavily depends on the symmetry (thanks to the assumptions), which may not hold in general cases. If considering weaker assumptions, I guess we may still prove that zero is a local minimum, since the regularization is quadratic and the square loss has higher order, but the quantitative results in this paper may not be very helpful.
> - First, while having too many symmetries may make the problem simpler, we stress that our result has much fewer symmetries than the theoretical standard of this problem. For example, in Kawaguchi (2016), there is no weight decay or stochasticity, which means the model in Kawaguchi (2016) is invariant to any invertible linear transformation. In comparison to Mianjy and Arora (2019), there is no weight decay, and so they contain rescaling symmetries in addition to ours. Thus, our result is more difficult to obtain than the previous works and significantly advances the theoretical frontier of this problem.
> - Secondly, there are two ways our theoretical result can be relevant to a more realistic setting with standard linearities. In ReLU networks, the rescaling symmetry of linear networks is retained, and so the balancing conditions such as Lemma 1 and Lemma 2 still hold for ReLU networks. For Tanh networks, the activation function is locally linear when the model is close to zero, and so our result can be seen as a perturbative approximation to the local landscape of a Tanh network, and this is why in situations such as in the right panel Figure 1, our result is relevant. Also, see the reply to the next question to see why our result is relevant to a ResNet.
>
> > Since the implications in Section 5 are all in general settings, the theoretical results may not support the discussions here strongly enough. In addition, I guess a ResNet architecture may avoid the local minimum at zero since the square loss is not in high order now.
> - Thanks for mentioning ResNet. In fact, our results are also to the standard ResNets in use, and we added additional experimental results to check. Using ResNet, one often needs to change the dimension of the hidden layer after every bottleneck, and a learnable linear transformation is often applied here. Thus, the “effective depth” of a ResNet would be somewhere between the number of its bottlenecks and its total number of blocks. For example, for a ResNet18 applied to CIFAR10, it often has five bottlenecks and 18 layers in total. We thus expect it to have qualitatively similar behavior to a deep linear net with a depth in between. See the updated Section 5 and Figure 2.
> We have also softened the tone of the implication section to make it clear to the reader that the implications should be taken with the caveat that, strictly speaking, our result only applies to linear nets.
> - We also added empirical results that shows that the landscape of linear nets can be either qualitatively or quantitatively similar to those of nonlinear nets (ReLU, tanh, Swish) close the origin [section G]
>
> For the specific questions:
> > An important result of this paper is that zero is a saddle point (not local minimum) when $D=1$ if the strength of the signal is more significant. It would be better to mention it in Theorem 1 or its discussion.
> - Thanks. We have updated the manuscript to reflect this.
>
> > In Figure 1 Left, the labels of the inequality seems need update.
> - We have updated the figure.
>
> > In Figure 2 Right, does the linear regression also have l2 regularization?
> - Yes. In fact, a linear regressor never has a trivial regime, however strong the L2 regularization is.

---

### Author Response · Authors · 2022-08-02
**Summary of revision**

We thank all the reviewers for the detailed and constructive assessment they provided. We have significantly extended our original theoretical and empirical results to answer the questions and improve the manuscript. We have also updated the manuscript to remove typos and overclaims. The major additions to the manuscript are colored in orange.

Specifically, we added
1. Theoretical results on the effect of including the bias term explicitly [section C]
2. Theoretical results on the case of zero weight decay ($\gamma=0$) [section D]
3. Theoretical results on the case of zero noise ($\sigma^2=0$) [section E]
4. Empirical results that relate deep linear nets to ResNet18 on CIFAR10 [Section 5 and Figure 2]
5. Empirical results that show that the landscape of linear nets can be either qualitatively or quantitatively similar to those of nonlinear nets (ReLU, tanh, Swish) close to the origin [section G]

With these additions, we are confident that the manuscript has been significantly improved, and the existing claims are much better supported with evidence. Also, if the reviewers point out any remaining problems, we are willing to make revisions to the manuscript accordingly. Please also see the specific replies to each reviewer.

---

### Meta-Review · Area_Chair_2f9d · 2022-08-26

**Recommendation:** Accept
**Confidence:** Less certain

**Metareview:**

There is a clear consensus amongst the reviewers that the manuscript advances the theory for linear deep networks to a degree warranting acceptance at NeurIPS.  The authors responded well to the issues raised by the reviewers which results in increased support by the reviewers that the manuscript be accepted.  Inclusion of weight decay, stochasticity, and architectures beyond feed forward networks make this a valuable addition to the theory of linear deep networks.

**Award:**

No

---

### Decision · Program_Chairs · 2022-09-14

Accept